# Mapping and modeling human colorectal carcinoma interactions with the tumor microenvironment

Ning Li [1,2,12] ✉, Qin Zhu[3,4,12], Yuhua Tian[1], Kyung Jin Ahn[4], Xin Wang[1], Zvi Cramer[1], Justine Jou[5], Ian W. Folkert[6], Pengfei Yu[6], Stephanie Adams-Tzivelekidis[1], Priyanka Sehgal[7], Najia N. Mahmoud[8], Cary B. Aarons[8], Robert E. Roses[9], Andrei Thomas-Tikhonenko [4,6,7,10], Emma E. Furth[6], Ben Z. Stanger [2,5,10], Anil Rustgi[11], Malay Haldar[6,10], Bryson W. Katona[5,10], Kai Tan[2,3,4] ✉ & Christopher J. Lengner [1,2,10] ✉

The initiation and progression of cancer are intricately linked to the tumor microenvironment (TME). Understanding the function of specific cancer-TME interactions poses a major challenge due in part to the complexity of the in vivo microenvironment. Here we predict cancer-TME interactions from single cell transcriptomic maps of both human colorectal cancers (CRCs) and mouse CRC models, ask how these interactions are altered in human tumor organoid (tumoroid) cultures, and functionally recapitulate human myeloid-carcinoma interactions in vitro. Tumoroid cultures suppress gene expression programs involved in inflammation and immune cell migration, providing a reductive platform for re-establishing carcinoma-immune cell interactions in vitro. Introduction of human monocyte-derived macrophages into tumoroid cultures instructs macrophages to acquire immunosuppressive and pro-tumorigenic gene expression programs similar to those observed in vivo. This includes hallmark induction of *SPP1*, encoding Osteopontin, an extracellular CD44 ligand with established oncogenic effects. Taken together, these findings offer a framework for understanding CRC-TME interactions and provide a reductionist tool for modeling specific aspects of these interactions.

Colorectal cancer (CRC) is the third most deadly and fourth most diagnosed cancer globally, with increasing incidence in both developing and developed nations and only minor gains in decreasing mortality rates, primarily among older patients (65+ years)[1,2]. While the reasons underlying the global burden and lack of major therapeutic advances in CRC are complex and multifactorial, a paucity of clinically relevant mouse and human models plays a role. Mouse genetic models have shed light on the molecular events leading to CRC initiation but are less tractable for modeling aggressive stages of the disease due to the number of driver gene mutations required to establish invasive adenocarcinoma as well as premature mortality resulting from obstruction

prior to tumor invasion and metastatic spread. In contrast, human CRC cell lines have been established from late-stage disease, but suffer from years-long culture adaptation, genetic drift, and absence of the complex in vivo microenvironment. In the past decade, breakthroughs in our understanding of the intestinal stem cell compartment have enabled, for the first time, the establishment of intestinal epithelial organoid cultures that retain stem cell function and karyotypic normalcy for long periods of time in vitro[3]. This knowledge subsequently enabled the establishment of organoids from resected human colorectal cancers[4].

As with their non-transformed counterparts, colorectal cancer organoids (hereafter referred to as tumoroids) maintain features of

tumor tissue architecture observed in vivo and, importantly, can predict response to radiation and chemotherapy treatments[5–10]. However, as with traditional 2D cancer cell lines, the lack of tumor microenvironment (TME) components in patient-derived tumor models precludes the development or evaluation of emerging TME-targeted therapies, such as those modulating the activity of cancer-associated fibroblasts or cells of the immune system. Recent advances in air-liquid-interface tumor cultures have begun to address this limitation, enabling modeling of immune checkpoint blockade, for example[11]. However, the TME components in tumor air-liquid-interface culture are present only transiently upon culture establishment[11]. Similarly, several studies have sought to understand epithelial-immune crosstalk using organoid co-cultures, including the effects of macrophages on intestinal barrier fidelity, and epithelial-T cell crosstalk[12–14]. Thus, gaining a holistic understanding of carcinoma-TME crosstalk in vivo and generating a framework for interrogating specific aspects of this crosstalk is critical for the development and evaluation of therapies aimed at altering this crosstalk.

Recently, several single cell transcriptomic profiles of human CRC have shed light onto the identity of numerous cell populations within the TME[15–20]. Using clustering analysis and trajectory inferences, several tumor-associated macrophage (TAM) populations were identified and postulated to derive from tumor-infiltrating monocyte precursors. These studies hypothesize that, upon association with the TME, macrophages acquire specific states characterized by hallmark expression of genes such as *SPP1* and *C1QC*, and, importantly, do not conform to classical models of M1/M2 macrophage polarization and rather appear to exist as a continuum of states. These studies hypothesized that TAM states are influenced by numerous factors within the TME, including immune cells, cancer-associated fibroblasts (CAFs), oxygen tension, and nutrient availability. However, functional confirmation of these hypotheses and insight into the potentially instructive role of carcinoma cells themselves is limited, in part due to the paucity of carcinoma cells in the datasets, as well as the absence of a reductionist experimental model in which the influence of carcinoma cells on macrophage precursors or other cells of the TME can be directly tested.

In this work, we sought to map putative interactions between human CRC cells and cells of the TME using single cell transcriptomic analyses of treatment-naïve, surgically resected human colorectal adenocarcinomas. In parallel, we established CRC tumoroids and colon organoids (colonoids), the latter from histologically normal adjacent epithelium. We asked how the selective pressure of ex vivo culture alters the carcinoma transcriptome and found a significant suppression of gene expression programs related to carcinoma-TME communication in culture. Using the tumoroid model as a framework for understanding specific carcinoma-TME interactions, we asked whether communication between human macrophages, cancer-associated fibroblasts (CAFs), and carcinoma cells could be re-established upon co-culture with tumoroids, given the abundance of these cells within the TME and their important roles in both tumor suppression and promotion. Remarkably, using this approach we found that interactions with carcinoma cells themselves are sufficient to instruct macrophages to induce pro-tumorigenic TAM identities and gene expression programs, including activation of a hallmark *SPP1*+ state. In contrast, CAFs alone are insufficient for polarization of SPP1+ macrophages, although the presence of CAFs in cocultures with tumoroids enhances the SPP1+ macrophage population. The SPP1+ state acquired by macrophages co-cultured with tumoroids is highly analogous to the SPP1+ state observed in tumor-associated macrophages in vivo, relative to macrophages derived from normal adjacent tissue. This SPP1+ macrophage state is associated with tumor immunosuppression and poor prognosis[21,22], and *SPP1* itself encodes Osteopontin, a well-established pro-oncogenic extracellular matrix (ECM) component and CD44 ligand capable of blunting T cell

activation[23]. Indeed, our analysis demonstrates strong correlations between SPP1+ macrophage states in vivo and immunosuppressive states within the adaptive immune compartment. Taken together, our findings highlight the limitations of carcinoma tumoroid culture models yet demonstrate that these limitations can be advantageous in reducing a highly complex system in order to functionally interrogate its specific components. Further, we conclude that carcinoma cells themselves are powerful and underappreciated regulators of TAM identity with the ability to induce pro-oncogenic, immunosuppressive states in human monocyte-derived macrophages.

## Results

### Single cell transcriptomics predicts extensive crosstalk between carcinoma cells and the tumor microenvironment

To begin understanding carcinoma-TME interactions in colorectal cancer, we set out to collect primary tumors, phenotypically normal adjacent colonic tissue, and liver metastases (in rare instances where metastases were surgically resectable concomitant to primary tumor resection). Surgical samples were then processed for single cell transcriptome profiling (scRNA-seq) and organoid culture in parallel (Fig. 1A). *In toto*, we collected tumors from 16 patients spanning tumor grade, stage, and CMS type[24] (Fig. 1B–D and Supplementary Fig. 1). The majority (14/16) of these tumors were microsatellite stable (MSS), with two exhibiting microsatellite instability (MSI) (Fig. 1B). Organoid cultures from tumor (tumoroid) and adjacent normal epithelium (colonoid) were maintained for 4-6 passages over a minimum of 2 months, and then subjected to scRNA-seq using methodology and reagents analogous to those used for primary tissue (Fig. 1A, C, D and Supplementary Fig. 1). We captured a total of 38,063 cells from the primary tumors, 11,221 cells from normal adjacent tissues, 5906 cells from 4 metastatic lesions, and 24,156/20,855 cells from in vitro cultured normal organoids and tumoroids, respectively. Cell type assignment based on scRNA-seq profiles indicated that, as expected, primary samples contained cells of the TME (non-epithelial), while organoid/tumoroid cultures consisted exclusively of epithelial/carcinoma cells (Fig. 1E).

To begin understanding carcinoma-TME interactions in colorectal cancer, we initially evaluated TME composition. Primary tumors contained a variety of cell types, including immune components (macrophages, dendritic cells, T-cells, B-cells, plasma cells, and mast cells), as well as non-immune cell types including endothelial cells, fibroblasts, and myofibroblasts with differing frequencies across patients (Fig. 2A, B, Supplementary Fig. 2A). Several minor clusters within the tumor epithelial cell population suggest heterogeneous expression patterns in carcinoma cells across patients, likely a reflection of varied mutational landscapes. (Supplementary Fig. 2A, F, G). Immune cells, especially macrophages, T cells, and Plasma/B cells, were present in high abundance in most patients (Fig. 2B). To understand epithelial-microenvironmental interactions spatially, we also generated cell type annotations in histological sections using CODEX (CO-Detection by indEXing[25]) spatial proteomics (Fig. 2C, D and Supplementary Fig. 2B–E). Unsurprisingly, CODEX analyses revealed that in both normal adjacent colon and tumor sections, epithelial (or carcinoma) cells were most likely to make homotypic contacts. However, carcinoma cells had a higher propensity for interaction with cells from the microenvironment relative to their normal counterparts, likely due to a breakdown of normal tissue architecture and tumor infiltration with stromal and immune components (Fig. 2D).

To globally map molecular crosstalk between these diverse cell types, we performed cell-cell communication analyses (see Methods), revealing extensive potential receptor-ligand interactions between carcinoma cells and cells within their microenvironment (Fig. 2E, F, Supplementary Data 1). Expanding this analysis into metastatic lesions, we also predict numerous carcinoma-TME interactions (Supplementary Fig. 3A–C, Supplementary Data 1). Interestingly, carcinoma cells

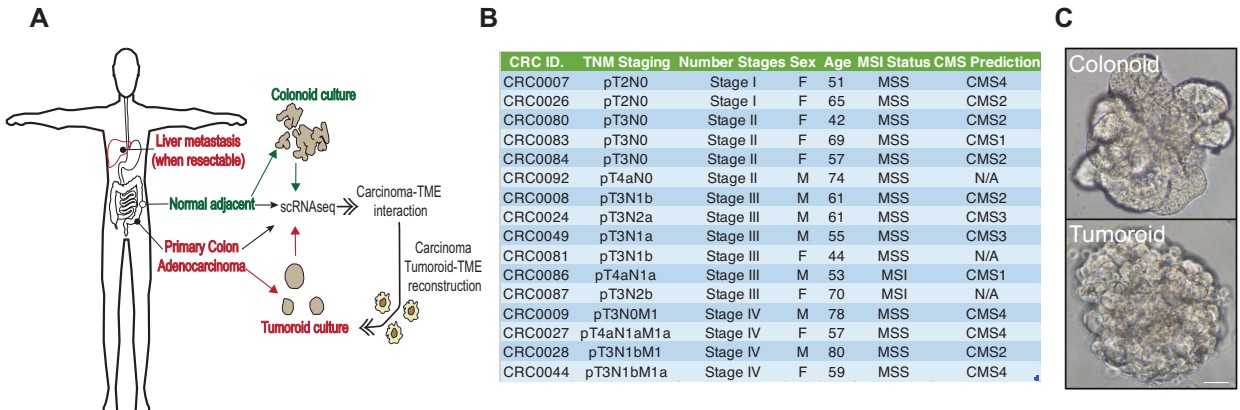

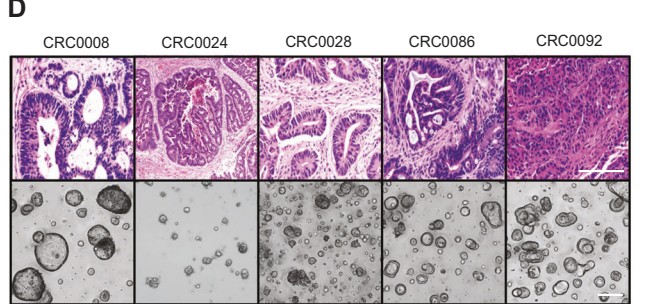

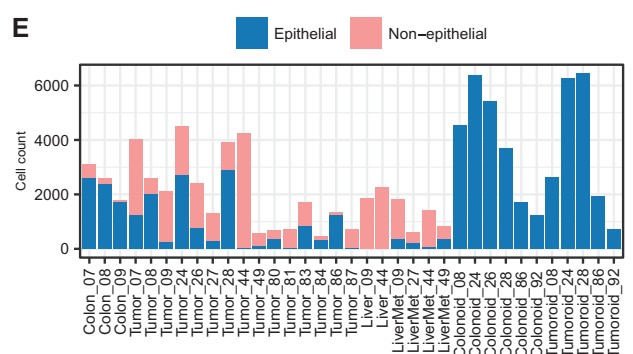

**Fig. 1 | Overview of experimental design: establishment of tumor organoid and single cell transcriptomic datasets. A** Treatment-naïve colorectal adenocarcinomas, liver metastases (when resectable concomitant to primary tumor resection) and normal adjacent colon was subjected to single cell transcriptomic profiling (scRNA-Seq). In parallel, primary tumor and normal adjacent colon samples were also used to seed tumoroid and organoid cultures which were subjected to scRNA-Seq after culture adaptation. To model TME-carcinoma interactions, healthy human donor-derived monocytes were differentiated into macrophages and co-cultured with organoids or tumoroids and subjected to scRNA-Seq. **B** Table providing patient/tumor data. MSI/MSS= Microsatellite instable/stable. TNM staging

Tumor/Node/Metastasis staging. **C** Representative brightfield, whole-mount micrograph of normal adjacent-derived colon organoids and primary tumor-derived tumoroids (scale = 50 µm). Organoids are passaged 4-6 generations prior to any analyses. **D** Representative hematoxylin/eosin histology for a subset of primary tumors and brightfield micrographs of their cognate 3D tumoroid cultures (scale = 100 µm). An area measuring 1.5 mm by 1.5 mm on the H&E slides was scanned, and a portion of that tile scan is displayed here. **E** Bar graph showing total sequenced cell count of each sample in scRNA-Seq datasets, stratified by coarse cell type epithelial (normal or carcinoma) and non-epithelial (normal stroma or TME). Source data are provided as a Source Data file.

that metastasize to the liver exhibited similar likelihoods of contact with TME cells as their counterparts in the primary tumor, with some exceptions (Supplementary Fig. 3A). Examining gene expression in metastatic carcinoma cells relative to their primary tumor counterparts, we were able to detect changes in the expression of several genes and pathways in metastatic lesions relative to their primary counterparts (Supplementary Fig. 3D, E). This includes the upregulation of pathways governing post-translational control of the canonical Wnt signaling pathway in metastatic carcinoma cells (Supplementary Fig. 3E). Canonical Wnt signaling is the central agonist of colon stem cell self-renewal[26], and thus this observation may reflect a selection for cancer stem-like cells in metastatic lesions, as has been observed in mouse models of metastatic colorectal cancer[27], however the small number of paired primary tumor-metastasis samples in our dataset ($n = 4$ pairs) warrants caution in generalization of these observations.

### Tumoroid culture alters cell type distribution and suppresses gene expression programs associated with epithelial-immune crosstalk

We next sought to understand how removal of carcinoma cells from their in vivo tumor environment and introduction into 3D organoid culture alters their gene expression programs, particularly as it relates to communication with the TME. We initially focused on

understanding the identity of epithelial/carcinoma cells derived from primary tissue versus long-term organoid culture. UMAP-based visualization of single cell transcriptomes and Pearson correlation reveal distinctions not only between normal epithelium and carcinoma cells, but also between primary and cultured cells (Fig. 3A, B). There was more concordance in transcriptional identity between normal epithelial samples across patients, both in primary colon and colonoid samples, and less concordance amongst tumor/tumoroid samples (Fig. 3B). Strikingly, the average transcriptome of cells from tumoroids and colonoids are more similar to each other than to their in vivo counterparts (Fig. 3B). To ask what might account for this, we examined heterogeneity within samples, which revealed a reduction in diversity as carcinoma cells are removed from the primary tumor and maintained in tumoroid culture (Fig. 3C). We then examined the fraction of cells with transcriptional identities more similar to crypt base columnar stem cells (SC), transit-amplifying progenitor cells (TA), or mature absorptive colonocytes (CC) across the four sample types (primary colon, organoids, primary tumor, tumoroids) (Fig. 3D, Supplementary Fig. 4A, B). As expected, we found that primary normal adjacent colon is enriched in differentiated colonocytes relative to normal colonoids, which have greater stem cell and transit-amplifying populations. Interestingly, tumoroids also shift towards stem cell and transit-amplifying identity relative to their in vivo counterparts

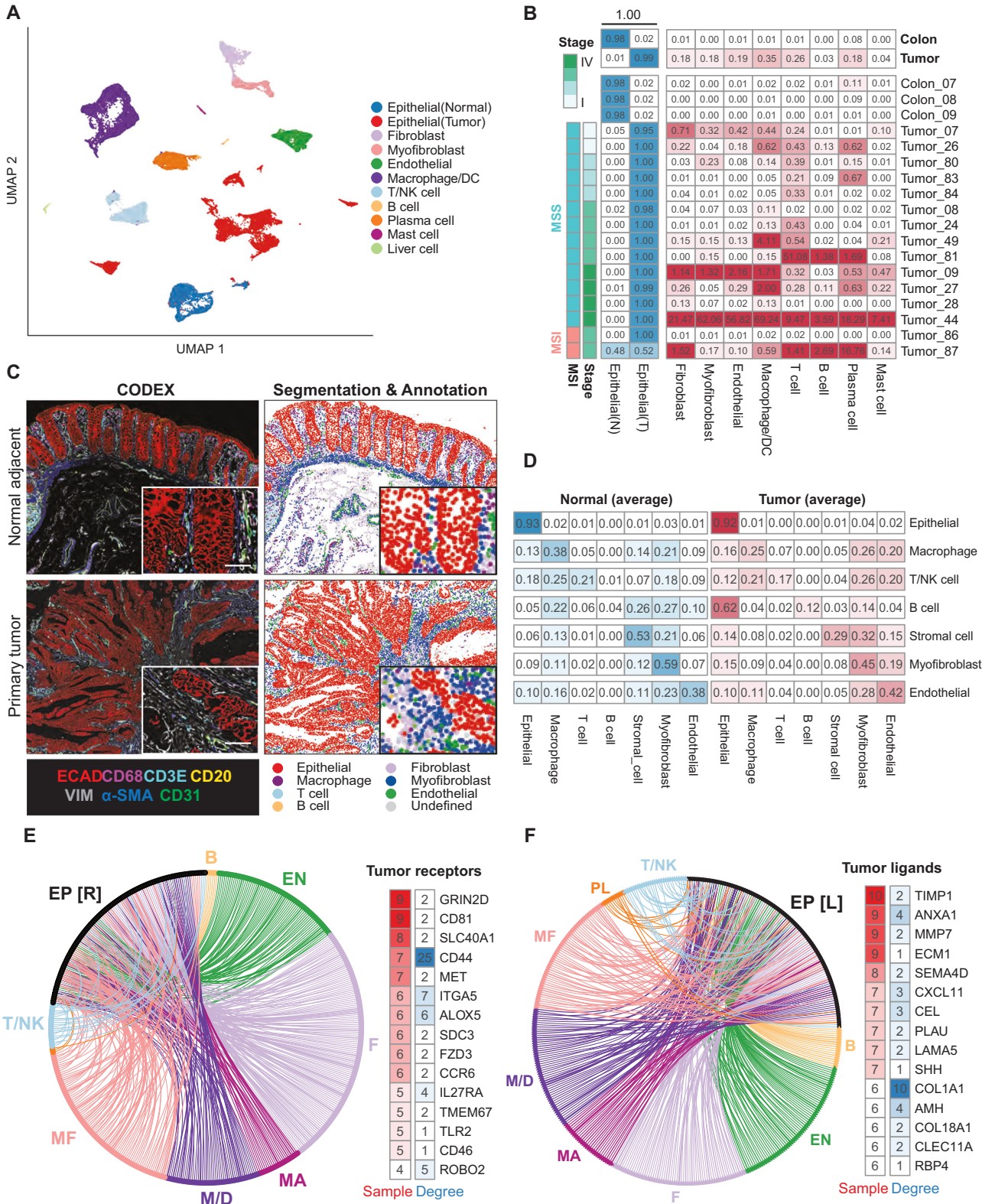

(Fig. 3D). These observations may be due to the nutrient- and niche cytokine-replete culture conditions driving increased stem cell self-renewal and proliferation relative to the in vivo environment.

Focusing on carcinoma cells, we next asked how the constraints of the tumoroid culture influence transcriptional identity. We found a greater number of genes suppressed vs. activated in tumoroid culture vs. primary carcinoma cells (Fig. 3E, Supplementary Data 2), as well as in colonoid culture vs. normal adjacent epithelium (Supplementary

Fig. 4C, Supplementary Data 2). Pathway analysis of these down-regulated genes indicates that tumoroid/colonoid culture primarily suppresses gene expression programs involved in communication with the immune system, particularly those related to leukocyte migration and inflammation (Fig. 3F, Supplementary Fig. 4D). This finding is consistent with the absence of immune cells in these long-term cultures (Fig. 1D). Using normal adjacent colon and colonoids as a control baseline, we compared the log2 fold change of gene expression in vitro

**Fig. 2 | Mapping carcinoma-TME interactions in human colorectal cancer.**
**A** Uniform Manifold Approximation Projection (UMAP) single cell transcriptomes from all cells collected in vivo, with cell types indicated. **B** Cell type composition in primary tumor and normal adjacent samples. Cell counts of each microenvironment cell type are normalized by total number of epithelial cells. Tumor samples are ordered based on MSI/MSS status and tumor stage. **C** Left panel: CODEX image of normal adjacent tissues and primary tumor from patient 86, highlighting seven cell-type markers – ECAD, CD68, CD3E, CD20, VIM, α-SMA and CD31 (scale = 50 μm). Right panel: post-segmented image colored by cell type. A section approximately 1.5 mm by 1.5 mm in size from the H&E slides was scanned, the exact dimensions of which vary based on the patient tissue acquired. A segment of this tile scan is presented here. **D** Composition of cell neighborhood based on CODEX image analysis. Each row represents the average cell type composition of a k-nearest-neighbors (k = 6) surrounding a particular cell type and the values sum to 1.

**E** Receptor-ligand interactions up-regulated in primary tumors. Each edge indicates a predicted interaction between a receptor upregulated in primary tumor carcinoma cells compared to normal adjacent epithelial cells, and a ligand expressed by indicated cell type in the TME. Edge widths indicate the number of patients (samples) in which the receptor is significantly up regulated. Receptors were ranked based on the number of patients with increased expression and the graph degree, which represents the number of ligands from each cell type that communicate with the receptor. EP [R] receptors expressed on epithelial cells, B B cells, T T cells, EN endothelial cells, F fibroblasts, MF Myofibroblasts, M/D Macrophage/dendritic cells, MA mast cells. **F** Same as E, but highlighting the ligands up-regulated in primary tumor carcinoma cells compared to normal adjacent epithelial cells, and corresponding receptors expressed by TME cells. EP [L] ligands expressed on epithelial cells, PL plasma B cells. Source data are provided as a Source Data file.

and in vivo and found many genes that are up-regulated in carcinoma cells in vivo are also up-regulated in vitro ($N = 7925$ in the first quadrant of Fig. 3G). However, the difference between tumoroids and colonoids was much smaller in vitro than that of their counterparts in vivo (fitted linear coefficient = 0.31 < 1, Fig. 3G), suggesting an overall repression of tumor-specific gene expression programs in culture. Interestingly, we found the repression of receptor and ligand expression to be greater than average ($p$-value = 0.024), indicating a significant impact on gene programs associated with cell-cell communication when moving into organoid culture systems (Fig. 3G), Conversely, gene expression programs related to cell division, patterning, and metabolism were generally activated in culture relative to in vivo tissue, consistent with the observed shift to stem and progenitor cell states and away from terminally differentiated absorptive states in culture (Fig. 3D, F and Supplementary Fig. 4D, E). Despite these shifts in the balance of cell type distribution upon in vitro culture, tumoroids and colonoids retain the diversity of epithelial cell types and cell type identity observed in the in vivo setting, indicating that culture does not inherently alter epithelial cell identity (Supplementary Fig. 5).

## Mapping extracellular matrix interactions and their changes in vivo and in vitro
Beyond the immune system, another key component of the TME is the extracellular matrix (ECM). We thus asked if pathways associated with ECM interaction are altered in the epithelial compartment. In the tumor vs. tumoroid comparison described in Fig. 3, we found gene ontology (GO) related to ECM organization to be significantly downregulated in tumoroid carcinoma cells relative to carcinoma cells in vivo (adjusted $p$-value = 3.5e−08, Supplementary Data 3). Analyses between tumor vs. colon, and tumoroid vs. colonoid indicates that genes associated with ECM organization are highly upregulated in carcinoma cells compared to normal epithelial cells (Supplementary Fig. 6A, B). However, there are relatively few differentially expressed genes (DEGs) related to ECM remodeling when comparing tumoroids vs. colonoids, suggesting the primary driver of gene expression differences related to extracellular matrix crosstalk is between in vivo and in vitro states, likely due to a switch from the endogenous ECM to Matrigel®.

To understand the differences in ECM-cell interaction between carcinoma and normal epithelial cells, and between in vivo and in vitro settings in more detail, we performed network analysis based on a curated ECM network database from matrixDB[28], obtained from the matrinetR package[29]. We computed pairwise gene expression correlations among carcinoma cells in vivo or from tumoroids and plotted the correlation as edge color on the ECM network (Supplementary Fig. 6C, D). The overall correlations between the ECM genes are higher in tumors in vivo compared to tumoroids (Supplementary Fig. 6E). Consistent with the GO analysis, the differences in the expression of multiple genes in the ECM network, including collagens (*COL1A1, COL1A2, COL3A1, COL4A1*), fibronectin (*FN1*), Lumican (*LUM*) and Osteonectin (*SPARC*), are highly significant in vivo compared to in vitro.

Fibroblasts, which include myofibroblasts (Supplementary Fig. 6F), are the major producers of ECM, and they were enriched in tumor samples (Fig. 2B, Supplementary Fig. 6G). Importantly, beyond the proportional differences in these populations, we found several genes upregulated in CAFs compared to normal fibroblasts. These include *CTHRC1, INHBA, BGN*, and *PDPN*, all of which are known to promote tumor progression[30–33].

## Relating human tumors, tumoroids, and mouse models of CRC
Given the importance of the TME, and particularly the immune system in colorectal cancer initiation and progression, we wondered how well human tumoroid models represent human primary tumors relative to common in vivo models of colorectal cancer in mice with intact immune systems. To this end, we asked how well various mouse CRC models recapitulate human primary tumors relative to human tumoroid models. We include the $Apc^{min}$ model of familial adenomatous polyposis, the AOM-DSS model of inflammation-driven colorectal cancer, and a model of invasive, metastatic colorectal adenocarcinoma generated by endoscope-guided orthotopic implantation of CRISPR/Cas9-engineered tumoroids with oncogenic *Apc, Trp53, Kras*, and *Smad4* (APKS) mutations into the colonic mucosa of syngeneic mice. Like human primary tumors, we observed the cancer cell populations shift towards a stem cell-like state in all three mouse models (Supplementary Fig. 4F–H). However, unlike in human, we did not observe significant population shift towards the proliferative transient-amplifying cell state in the mouse tumors (Supplementary Fig. 4H). Remarkably, human tumoroid-specific transcriptomes are better correlated with human primary-tumor-specific transcriptomes relative to mouse models, despite the absence of the TME in culture (Supplementary Fig. 4I, J). Taken together, these findings indicate that the primary effect of long-term tumor organoid culture is a loss of gene expression programs involved in communication with cells of the tumor microenvironment, and that human organoid models may offer advantages in modeling CRC not captured by commonly employed, immune-competent mouse models.

## Human tumors are enriched for pro-tumorigenic macrophage states and depleted of antigen-presenting and pro-inflammatory macrophage states
Recently, several studies have begun elucidating the identity and function of myeloid cells, and particularly macrophages, in the tumor microenvironment of human colorectal cancers[15,17]. Tumor-associated macrophages (TAMs) have been implicated in immune suppression, and macrophage depletion can, in certain tumor types, enhance tumor response to immune checkpoint blockaded[34]. TAMs exhibit unique transcriptional gene expression programs that are historically thought to be influenced primarily by the TME (e.g., nutrient availability, hypoxia, fibrotic matrix, CAFs, and other immune cell types)[22,34]. We initially examined the identity and distribution of myeloid populations within our in vivo-derived single cell transcriptomic data (Fig. 4A, B).

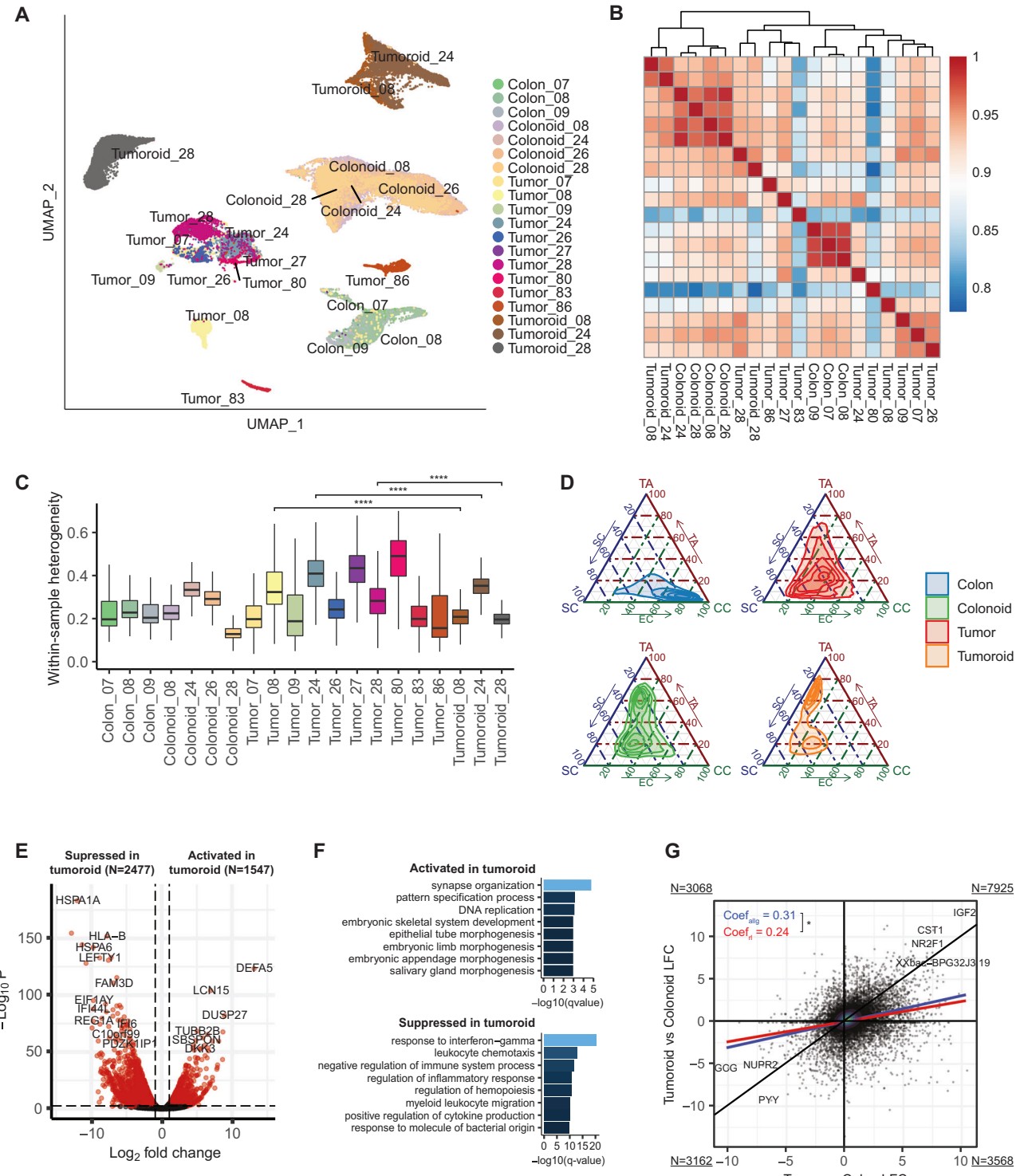

We identified a large population of macrophages, including those associated with tumors (primary or liver mets) or with normal adjacent tissue (normal colon or liver). We also identified a number of other myeloid dendritic cell (DC) types, including BATF3+ DCs required for effector T cell trafficking and adoptive T cell therapy[35], CD1c+ DCs which prime cytotoxic T cell responses[36], Tumor-associated LAMP3 + DCs[37,38], and LILRA4+ plasmacytoid dendritic cells[39].

We next sought to characterize the major macrophage sub-populations present. Using clustering and differential expression analysis (see Methods) we identified three major populations: those in an IL1B+ state, those in an SPP1+ state, and those in a C1QC+ state (Fig. 4C).

These states are characterized by the co-expression of groups of genes (Supplementary Fig. 7A) and are named for the expression of the hallmark genes *IL1B*, *SPP1*, and *C1QC*. Examining the distribution of these states in macrophages across sample types revealed that the C1QC+ state is most prevalent in normal adjacent colon tissue where it is associated with antigen presentation and modulation of the adaptive immune response (Fig. 4D–F, Supplementary Fig. 7B). The IL1B+ state is most prevalent in normal liver tissue and is characterized by inflammatory response signatures (Fig. 4D–F, Supplementary Fig. 7B). In contrast, the SPP1+ state is more prevalent in tumors relative to the other states, both in primary tumors vs. normal colon, in liver mets vs.

**Fig. 3 | Adaptation to organoid culture suppresses gene expression programs involved in carcinoma-TME communication. A** UMAP of epithelial cells colored by sample. Five tumor samples were excluded from this analysis due to low epithelial cell count. **B** Correlation heatmap showing between-sample similarity. Pearson correlations were calculated using average gene expression of epithelial cells for each sample pairs. **C** Within-sample heterogeneity across all samples. For each sample, average cosine distance was computed using 500 randomly sampled epithelial cells and displayed as boxplot. The height of the box ranges from the first quartile to the third quartile. The upper whisker indicates maxima but does not exceed 1.5 * IQR from third quartile. The lower whisker extends to the minima or 1.5 * IQR from the first quartile (IQR: inter-quartile range). All the box plots depicted in the subsequent figures adhere to this definition. One-sided student's $t$ test was performed for patients with paired tumor and tumoroid samples. ****$p \leq 0.0001$. **D** Ternary plot showing distribution of cells in the cell-type-signature space. Signature scores, derived from differential expression analysis on normal epithelial

cells, were computed with AUCell using signature genes and rescaled from 0 to 100. A 2D kernel density estimation visualized the distribution. **E** Volcano plot showing genes activated and suppressed in tumoroid compared to primary tumor based on differential expression analysis. Red point represents genes significantly up- or down-regulated in tumoroid compared to tumor, with FDR ≤ 0.01 and log2 fold change > 1. Adjusted $p$-values (FDRs) were displayed. **F** GO functional analysis of DEGs suppressed in tumoroid. $-\log_{10}(q\text{-value})$ of the top significantly enriched GO terms in biological process were plotted. **G** Scatter plot showing gene log fold change correlation in vivo (tumor vs. adjacent normal, x-axis) and in vitro (tumoroid vs. organoid, y-axis). Black line denotes y = x, where in vivo and in vitro differences are identical. Blue and red lines are linear regression fits for all genes (coefficient: 0.31) and for receptors and ligands (coefficient: 0.24), respectively. A one-sided Z test assessed coefficient equality with a $p$-value of 0.024. Quadrant corner numbers indicate gene counts. Source data are provided as a Source Data file.

normal liver, and in more advanced vs. early-stage tumors (Fig. 4F, Supplementary Fig. 7B). We predicted extensive receptor-ligand interactions between carcinoma cells and macrophages in each of these states (Supplementary Fig. 7C, D, Supplementary Data 1).

The SPP1+ state is characterized by glycolytic gene expression programs, responses to oxygen levels, and gene expression programs related to ECM organization (Fig. 4E). Importantly, SPP1+ macrophages have previously been associated with suppression of adaptive immune responses and thus this is considered a pro-tumorigenic state. Consistent with this, we find abundant SPP1+ macrophages in microsatellite-stable (MSS) tumors, generally considered more immunosuppressive tumors relative to the hypermutated, microsatellite-instable (MSI) tumors where macrophages preferentially reside in the inflammatory IL1B state (Fig. 4G).

Prior studies suggest that polarization of macrophages toward an SPP1+ state is a result of TME properties, including oxygen tension, the presence of FAP+ cancer-associated fibroblasts, and ECM composition, consistent with our pathway analysis[22,40] (Fig. 4E). Interestingly, *SPP1* itself encodes Osteopontin, a secreted ECM component and ligand for CD44 with known pro-tumorigenic and immunosuppressive activity[21–23,40–42]. Examining expression of *SPP1* and its receptor *CD44* in the TCGA dataset (COAD and READ) reveals concerted upregulation of this receptor-ligand pair in tumors relative to normal tissue (Supplementary Fig. 7E), whereas expression of *C1QC* is significantly lower in the TCGA tumors (Supplementary Fig. 7E). Consistent with these reported functions of SPP1 and macrophages in the SPP1+ state, stratification of colorectal cancer TCGA transcriptomes by SPP1 signature enrichment reveals significant reductions in patient survival when tumors have high SPP1 signature (Fig. 4H).

Given that these conclusions are drawn largely from single cell transcriptomic analyses, we sought supporting spatial evidence in histological sections of tumor and normal adjacent tissue using CODEX[25]. We observed macrophages present in histologically normal colonic epithelium were more likely to be HLA-DQA1+ (a proxy marker for the C1QC+ state, Supplementary Fig. 7A), while those present in tumor tissue were more likely to be SPP1+ (Fig. 4I–K). Further, single molecule fluorescent in situ hybridization confirmed the expression of *SPP1* in tumor-associated macrophages in CRC tissue (Fig. 4L). Together, these data demonstrate that tumor-associated macrophages are primarily in an SPP1+ immunosuppressive state (particularly in MSS tumors), and tumors suppress pro-inflammatory and antigen-presenting states.

## Adaptive immune cells and their relationships to macrophage states

Adaptive immune cells are crucial components of the tumor microenvironment, and there is great interest in understanding how tumors and the TME contribute to immunosuppression in 'cold' tumors. We therefore examined both T and B cell lineages in our datasets (Supplementary Figs. S8 and S9).

We identified major T/NK cell subpopulations including naive or memory T cells (Tn/Tm, CCR7 + ), regulatory T cells (Tregs, FOXP3 + ), effector T or effector memory T cells (Teff/Tem, GZMA/GZMB + ), short-lived effector cells (SLEC, KLRG1 + ), NKT cells (CD3 + NKG7 + ) as well as a population of cycling T cells (MKI67 + ), a cluster of T-carcinoma cell doublets and NK cells (Supplementary Fig. 8A, B). The distribution of these T/NK cell subpopulations differs across sample types and tumor stage (Supplementary Fig. 8C–E), with tumor and liver metastasis samples showing some commonality, despite the different anatomical location (Supplementary Fig. 8D).

Specifically, we observed enrichment of Tregs, Teff/Tem, and reduction of Tn/Tm and NK cells in tumor tissue relative to normal colon/liver (Supplementary Fig. 8D). There is also a trending increase in the proportion of Tregs and a decrease in the Teff/Tem population in late-stage patients (Supplementary Fig. 8E), indicative of a more immunosuppressive environment. For almost all T cell subpopulations, the exhaustion score, computed based on a set of known T cell exhaustion markers, exhibited significant elevation when comparing T cells in the TME against those in adjacent normal tissue (Supplementary Fig. 8F).

Lastly, we asked if there exists an association between the observed macrophage states and T cell states by looking at the correlation between the fraction of the T cells in different states in each sample, and the average IL1B/SPP1/C1QC state probability in macrophages. As shown in Fig. 4M, N and Supplementary Fig. 8G, the IL1B+ inflammatory macrophage state is highly correlated with the proportion of Tn/Tm and NK cells. In contrast, the SPP1+ state is positively correlated with the proportion of Treg, SLEC and NKT cell, and negatively correlated with the proportion of Tn/Tm, Teff/Tem, NK cell and cycling T cells, consistent with a role for SPP1+ macrophages, and Osteopontin itself, in tumor immunosuppression. C1QC+ state probability shows almost the opposite trend. The trend is even more striking if only tumor samples are included in the analysis (Fig. 4N, Supplementary Fig. 8G). These observations are consistent with the observations in our study and in the broader literature that the SPP1+ macrophage state is indicative of an immunosuppressive TME. It further indicates that T cell and macrophage states are highly coordinated, possibly due to the crosstalk between the two cell types.

A set of similar analyses were performed on the B/plasma cell subpopulations. We identified naïve B cells, cycling/differentiating B/plasma cells, and plasma cell subpopulations expressing different antibodies (Supplementary Fig. 9A, B). In the normal colon, B cells are mainly present in the form of *IGHA1*-expressing plasma cells involved in maintaining intestinal immunity[43]. In most cancer patients, there is a switch to *IGHG1*-expressing plasma cells (Supplementary Fig. 9B–D). When computing the correlation with macrophage state probability, suppression of B cell proliferation and differentiation was observed in samples with high SPP1+ macrophage signatures (Pearson correlation = −0.37), and the inverse trend was seen for the C1QC+ state

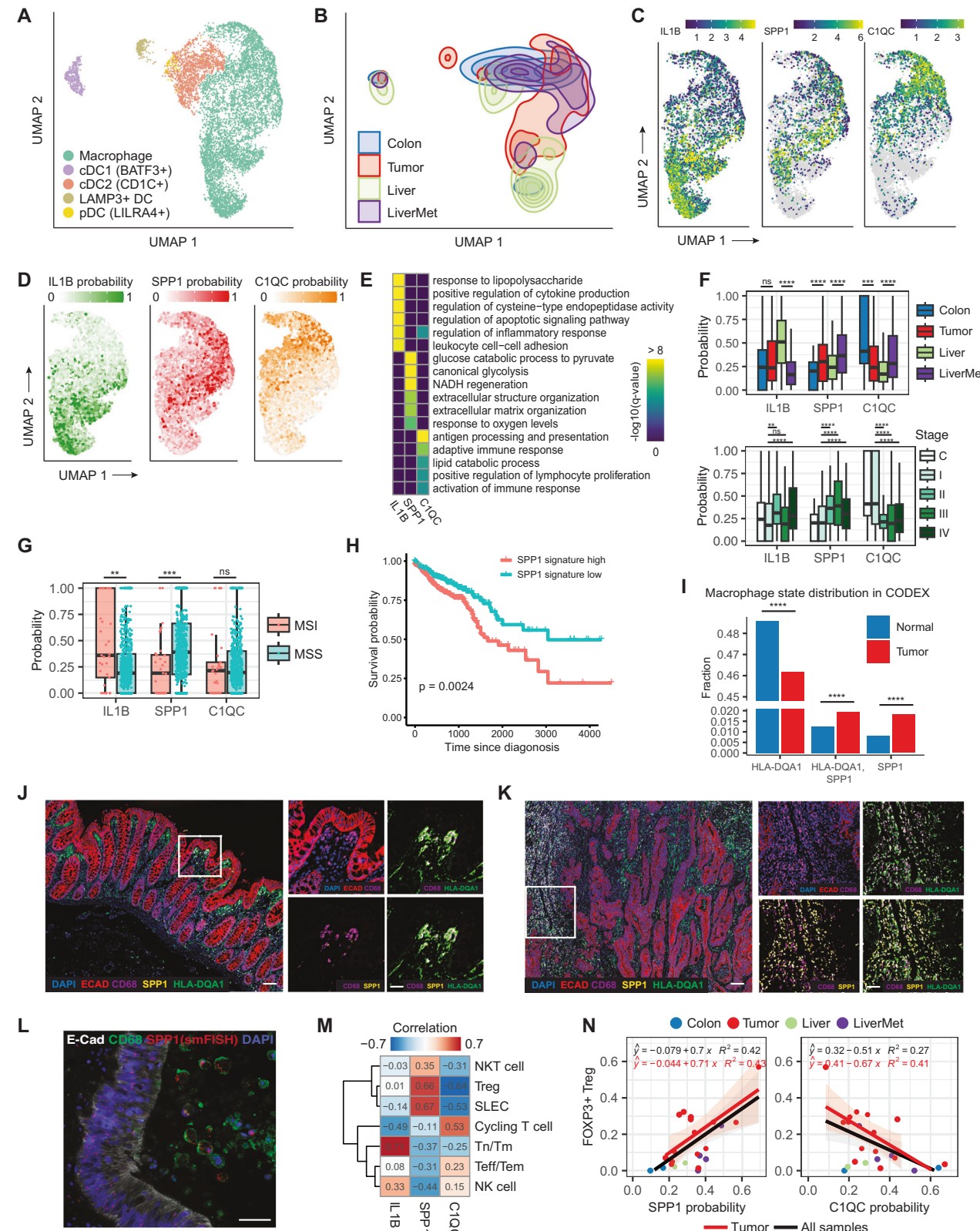

(Pearson correlation = 0.47), likely indicative of the different roles these macrophages play in the tumor TME (Supplementary Fig. 9B, E, F).

### Mouse models of colorectal cancer recapitulate macrophage state distributions observed in human tumors

Ultimately, these analyses in human tissue provide valuable insight into the carcinoma-immune TME interactions of colorectal cancer, but platforms for performing functional experiments are severely limited in

human systems and mouse models remain the primary platform for performing functional assays targeting carcinoma-TME interactions. We therefore returned to examine the three widely utilized mouse models of colorectal cancer introduced above - the *Apc^min/+* model of familial adenomatous polyposis, the AOM-DSS model of inflammation-driven colorectal cancer, and the endoscope-guided orthotopically implanted *APKS* tumoroid model of metastatic colorectal cancer (Fig. 5A). Similar to their human counterparts, these mouse models exhibit similar TME

**Fig. 4 | Myeloid cells and macrophage states in human CRC. A** UMAP of myeloid cells present in in vivo-derived single cell transcriptomic datasets. **B** UMAP as in **A**, but highlighting the distribution of cells across indicated sample types. **C** Expression of state-specific hallmark genes in macrophage population. **D** Probability of single cells transitioning to macrophage states, determined by the fateID algorithm. **E** Heatmap showing $-\log_{10}$($q$-value) of top enriched GO terms for each of the macrophage states. **F** Macrophage state probability distribution across sample types (top) with two-sided $t$-tests comparing colon to tumor, liver, and liver metastasis. Macrophage state probability across tumor stages (bottom) includes macrophages from adjacent normal colon samples (C colon) and $t$-tests between late-stage TAMs and stage I tumor TAMs. The height of the box ranges from the first quartile to the third quartile. The upper whisker indicates maxima but does not exceed 1.5 * IQR from third quartile. The lower whisker extends to the minima or 1.5 * IQR from the first quantile (IQR inter-quartile range). **G** Macrophage state probability between MSI and MSS samples. Test and significance labels as in **F**. The height of the box ranges from the first quartile to the third quartile. The upper whisker indicates maxima but does not exceed 1.5 * IQR from third quartile. The lower whisker extends to the minima or 1.5 * IQR from the first quantile (IQR inter-quartile range). **H** Survival analysis on patients stratified by gene signature score

(calculated using the AUCell package) of SPP1+ macrophage state. Survival analysis was carried out using the survival and survminer R package. **I** Fraction of macrophages in tumor and normal samples from patients 28, 86, and 92, as indicated by CODEX imaging. HLA-DQA1 marks C1QC+ state. A two-sided proportion test evaluated significant differences between tumor and normal samples. **J** Left: CODEX image of normal adjacent colon tissue from patient 28 (scale = 50 μm). Right panels: zoom-in images (scale = 25 μm). A section approximately 1.5 mm by 1.5 mm in size was scanned, and a segment of this tile scan is presented here. **K** Analogous CODEX analysis as in **J**, but looking at primary tumor tissue from patient 28 (scale = 50 μm). **L** CRC tumor section with smFISH for *SPP1* mRNA (red) co-immunostained for E-CADHERIN (white) and CD68 (green) protein. DAPI in blue (scale = 50 μm). **M** Correlation between the T cell subtype fraction and average macrophage state probability across all in vivo samples. **N** Association between Treg fraction and macrophage SPP1+ and C1QC+ state probability in in vivo samples. Linear regression was applied to all in vivo or just tumor samples. The band around the line represents the 95% confidence interval. In all panels, *$p \leq 0.05$; **$p \leq 0.01$; ***$p \leq 0.001$; ****$p \leq 0.0001$; ns: $p > 0.05$. Source data are provided as a Source Data file.

composition with significant myeloid cell infiltration relative to normal mouse colon (Fig. 5A, B). Mapping predicted receptor-ligand interactions in the mouse models and cross-referencing these interactions with those observed in humans reveals that the mouse tumors capture only about half of the receptors and ligands found in humans (Fig. 5C, D, Supplementary Data 1), which may be associated with the limit to DEG recovery in mouse models relative to human tumoroids when compared to primary human tumors seen in Supplementary Fig. 4J. Despite this apparent limitation, however, examination of mouse macrophages for the presence and distribution of states found in human tissue revealed that IL1B +, SPP1 +, and C1QC+ states could all be readily identified (Fig. 5E). Consistent with human tissue, SPP1+ states were enriched in all three mouse tumor models relative to normal colon, and C1QC+ states were suppressed in all three tumor models (Fig. 5F). Thus, while these mouse tumor models do not perfectly reflect the cell type distributions and putative carcinoma-TME communication we observe in human, they appear to largely recapitulate macrophage state changes associated with malignancy.

**Carcinoma cells instruct macrophages to enter pro-tumorigenic immunosuppressive states**

The TME has been implicated in dictating macrophage polarization within tumors, however, the degree to which carcinoma cells themselves directly influence macrophage identity remains largely unexplored, particularly in colorectal cancer. We therefore set out to address the degree to which carcinoma cells influence macrophage polarization by taking advantage of the tumoroid model system, free of TME components and absent the gene expression programs underlying TME-carcinoma crosstalk. To this end, we generated human monocyte-derived macrophages ex vivo via stimulation with human M-CSF, then introduced these macrophages into either normal organoid or tumoroid cultures, followed by single cell transcriptomic profiling of both the macrophage and epithelial components of the cultures (Fig. 6A and Supplementary Fig. 10). In response to introduction of macrophages into tumoroid cultures derived from patient 28, we observed a clear shift in macrophage polarization toward the SPP1+ state (Fig. 6B, C). This shift was consistently seen in macrophages co-cultured with tumoroids derived from five different patients (Fig. 6C and Supplementary Fig. 10). In some patient-derived cultures there was also modest induction of the IL1B+ state, concomitant to a suppression of the C1QC+ state, largely reflecting the in vivo states observed in tumor tissue vs. normal adjacent observed in Fig. 4 (Fig. 6C). Interestingly, carcinoma cells from patient 28 did not reciprocally respond to the presence of macrophages, suggesting that perhaps extended culture may result in epigenetic repression of transcriptional programs responding to the presence of TAMs, and/or the low levels of the

Osteopontin receptor CD44 in this tumoroid line (Fig. 6B, D). By contrast, tumoroid cultures from patients 86 and 92 were responsive to macrophage presence and expressed higher levels of CD44 relative to the non-responsive carcinoma cells from patient 28 (Fig. 6C–E).

Our observation that co-culture of macrophages with tumoroids is sufficient to induce macrophage polarization into the SPP1+ state raises several questions. Various other aspects of the TME have been implicated in macrophage polarization, including the presence of cancer-associated fibroblasts (CAFs)[16,22], and the relative contribution of carcinoma cells vs. CAFs in eliciting this response in macrophages is not clear, nor is the mechanism through which carcinoma cells elicit the SPP1+ polarization response. To address these questions, we performed additional co-culture assays with macrophages alone, macrophages in coculture with organoids/tumoroids, in coculture with CAFs, in coculture with CAFs and organoids/tumoroids, or in culture with organoid/tumoroid conditioned media (Supplementary Fig. 10A, B), followed by single cell transcriptome profiling. Here, using tumoroids and normal adjacent organoids from patients 8 and 24, we observe that CAFs have limited ability to induce SPP1+ macrophage polarization relative to carcinoma cells, and that the presence of both carcinoma cells and CAFs elicits the strongest SPP1+ macrophage polarization (Supplementary Fig. 10C–E). Interestingly, the application of tumoroid/organoid conditioned media to macrophage cultures was insufficient to induce SPP1 polarization, and rather induced polarization toward the C1QC+ immunogenic state, indicating that direct contact between macrophages and epithelial cells is required for SPP1 polarization (Supplementary Fig. 10C–E).

The clearest consequence of exposing macrophages to carcinoma cells culture (with or without CAFs) was polarization toward an SPP1+ state. The *SPP1* gene product Osteopontin is an extracellular matrix protein with pleiotropic signaling functions. While Osteopontin produced by TAMs contributes to blunting the adaptive immune response to cancer, it has a wide range of cancer cell-autonomous functions including the promotion of an epithelial-to-mesenchymal transition (EMT) linked to increased cancer stem cell properties and metastatic proclivity downstream of its interactions with CD44 and integrin $\alpha_v\beta_3$[42,44]. We therefore examined the nature of the response of carcinoma cells to the presence of macrophages in co-culture. Macrophage introduction resulted in varying degrees of differential gene expression in patient-derived tumoroids and normal adjacent organoids, and, interestingly, was generally associated with decreased cell cycling of carcinoma cells, but increased cycling of macrophages (Fig. 6F). This was concomitant to an increase in EMT signature gene expression in tumoroids and organoids upon co-culture with macrophages (Fig. 6G), with up-regulation of the mesenchymal marker Vimentin in patient 86 and 92 (Fig. 6H). This finding is consistent with a model whereby

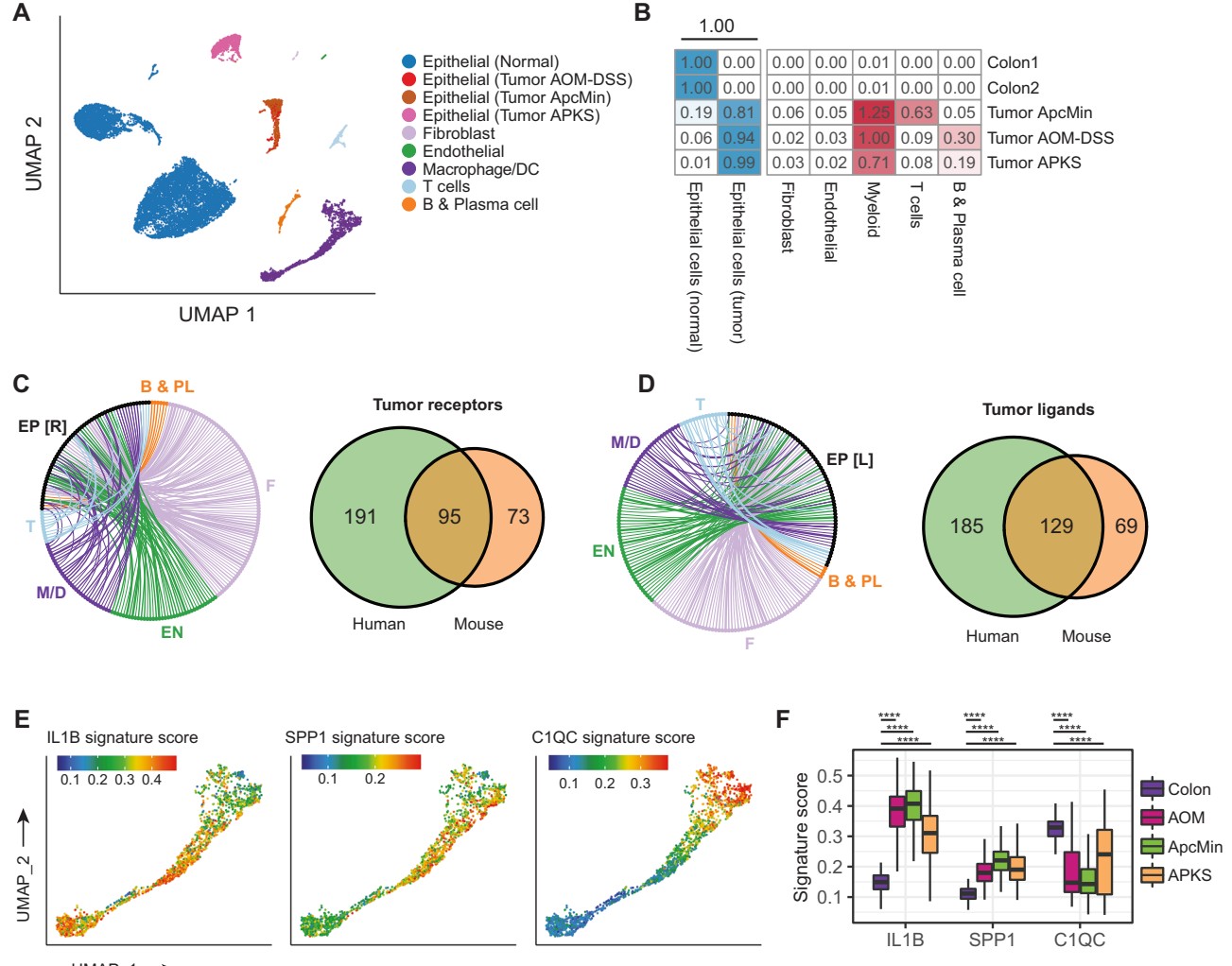

**Fig. 5 | Mouse colorectal cancer models recapitulate human tumor-associate macrophage states. A** UMAP of all cells from mouse CRC models, colored by cell types. **B** Cell type composition in mouse CRC models (AOM-DSS inflammation driven colorectal cancer, Apc<sup>min/+</sup> model of familial adenomatous polyposis, and APKS model employing endoscope-guided orthotopic implantation of engineered tumoroids into the colonic mucosa of immune-competent mice) and normal control colon. Cell counts for each microenvironmental cell type are normalized by total number of epithelial cells. **C** Receptor-ligand interactions up-regulated in mouse CRC models. Each edge indicates communication between a receptor upregulated in mouse tumor epithelial cells vs. normal colon epithelial cells, and a ligand expressed by TME cells. Edge widths indicate the number of CRC models in which the gene is differentially expressed. Venn diagram on the right indicate the overlap of tumor-specific receptors between mouse and human. EP [R] receptors expressed on epithelial cells, B & PL B and plasma B cells, T T cells, EN endothelial cells, M/D Macrophage/dendritic cells. **D** Same as **C**, but highlighting the ligands up-

regulated in mouse tumor epithelial cells compared to normal colon epithelial cells, and corresponding receptors expressed by TME cells. EP [L] ligands expressed on epithelial cells. **E** UMAP of macrophages colored by a score that measures the extent to which each single-cell transcriptome is enriched for genes specific to each of the macrophage states observed in human. Signature genes were derived by differential expression and were converted to mouse orthologs. Gene set enrichment scores were computed using the AUCell package[44]. **F** Distribution of signature scores for macrophage states in normal mouse colon and mouse CRC models. The height of the box ranges from the first quartile to the third quartile. The upper whisker indicates maxima but does not exceed 1.5 * IQR from third quartile. The lower whisker extends to the minima or 1.5 * IQR from the first quantile (IQR: interquartile range). Two-sided student's *t* test was performed between macrophages from normal colon and those from tumor models. Source data are provided as a Source Data file.

carcinoma-induced macrophage SPP1 promotes EMT, potentially providing a mechanistic basis for the previously observed link between SPP1 expression and colorectal cancer metastasis[17,44].

Ultimately, we asked to what extent introduction of macrophages into tumoroid cultures recapitulated the macrophage-carcinoma receptor-ligand crosstalk observed in vivo. Superimposing in vitro receptor-ligand interactions upon the full interactome between carcinoma cells and macrophages in vivo (Fig. 6I, light lines) reveals that a subset of these interactions are re-established in culture (Fig. 6I, dark lines), including the SPP1-CD44 interaction, among others. This predicted SPP1-CD44 interaction was validated in tumoroid-macrophage cocultures via proximity-ligation assays (Fig. 6J).

Taken together, this study demonstrates that human tumor-derived organoid culture suppresses transcriptional programs underlying crosstalk between carcinoma cells and the TME in vivo. However, this apparent limitation also offers an opportunity for focused reconstruction of specific carcinoma-TME interactions, enabling a reductionist approach to the functional evaluation of these interactions in what is otherwise a highly complex system in vivo.

## Discussion
The recent co-emergence of organoid culture systems and single cell transcriptomics offers unprecedented opportunities for understanding the complex communication between cancer cells and the

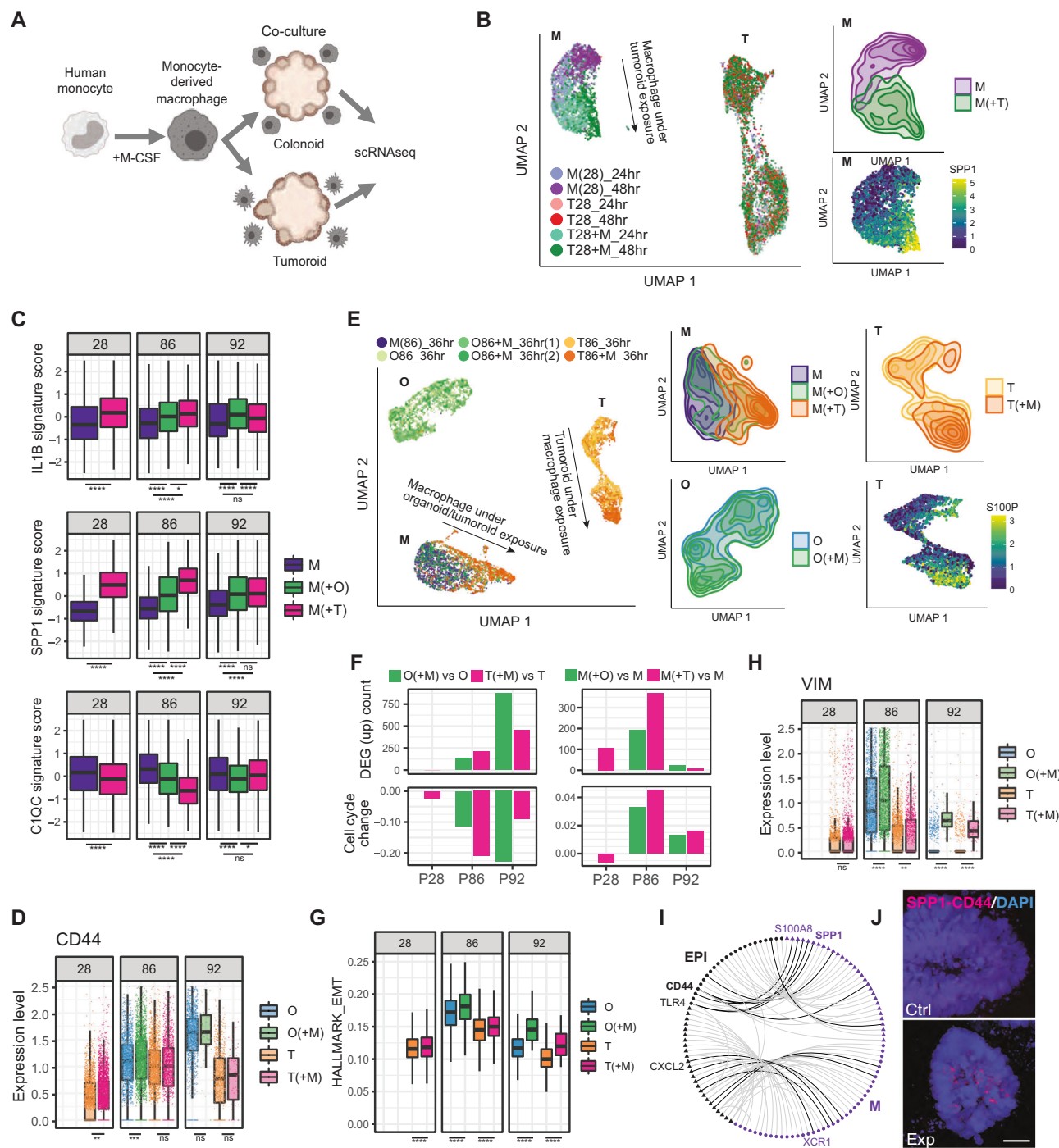

cells within their microenvironment. Here, we map putative carcinoma-TME interactions from scRNAseq data derived from treatment-naïve colorectal adenocarcinoma, ask how the gene expression programs underlying these interactions are altered in established tumor organoid cultures, and address whether these interactions can be re-established and functionally tested through introduction of specific TME components (macrophages) into the organoid cultures.

The advent of organoid culture, and subsequently tumor organoid (tumoroid) culture, holds promise for the advancement of patient-specific personalized medicine[45]. Recent studies indicate that patient-derived tumoroids can successfully predict response to radio- and chemotherapies[5–10], and tumoroids have been employed to study the development of tumor-reactive T cells[46] or the response to

immune checkpoint inhibitors in coculture systems[11]. In the context of colorectal cancer, air-liquid interface culture systems support survival of cells of the TME in conditions that favor carcinoma cell growth for several weeks, ultimately non-carcinoma cells do not self-renew and are lost over time[11].

We initially set out to ask how colorectal carcinoma cells and normal colonic epithelium respond to adaptation to the three-dimensional organoid culture. Our findings were not surprising: organoid culture favors a population shift towards proliferative stem and progenitors, likely resulting from the nutrient and growth-factor replete medium and non-hypoxic environment (our study is conducted using roughly physiological oxygen concentrations of 5%). Long-term organoid culture also led to suppression of gene expression programs involved in cell-cell communication (receptor-ligand pairs)

**Fig. 6 | Reconstructing carcinoma-macrophage interactions in tumoroid culture. A** Illustration of tumoroid/organoid-macrophage co-culture assay. **B** UMAP of tumoroid and macrophages before and after co-culture. The top right panel highlights the distribution of macrophage. The bottom right panel shows the expression of SPP1 in macrophages. M macrophages, T tumoroid cells. **C** Distribution of signature score for macrophage states before and after exposure to co-culture. Data are collected from three independent co-culture experiments with cells from different patients. The height of the box ranges from the first quartile to the third quartile. The upper whisker indicates maxima but does not exceed 1.5 * IQR from third quartile. The lower whisker extends to the minima or 1.5 * IQR from the first quantile (IQR: inter-quartile range). One-way ANOVA and Tukey's 'Honest Significant Difference' test was used to compare the average signature scores. **D** Boxplot showing the expression level of CD44 across condition and patients. The height of the box ranges from the first quartile to the third quartile. The upper whisker indicates maxima but does not exceed 1.5 * IQR from third quartile. The lower whisker extends to the minima or 1.5 * IQR from the first quantile (IQR: inter-quartile range). One-sided $t$ test was performed with the alternative hypothesis that CD44 expression increased upon co-culturing with macrophage. **E** UMAP of macrophages, tumoroid and organoid cells from Patient 86 before and after co-culture. First three panels on the right highlight the distribution of cell states of macrophages (M), tumoroids (T) and normal adjacent organoids (O) before and after co-culture. Last panel shows expression of S100P, a gene up-regulated in tumoroid upon co-culture. **F** Top left panel shows number of

significantly up-regulated genes in each organoid and tumoroid in response to the macrophages. Bottom left panel shows the change in the fraction of organoid/tumoroid cells entering S/G2/M phase upon co-culture. Right panels show the same set of statistics, but for macrophages before and after co-culture. **G** EMT signature score of organoids and tumoroids with or without macrophage co-culture. One-sided $t$ test was performed with the alternative hypothesis that EMT score becomes higher upon co-culturing with macrophage. The height of the box ranges from the first quartile to the third quartile. The upper whisker indicates maxima but does not exceed 1.5 * IQR from third quartile. The lower whisker extends to the minima or 1.5 * IQR from the first quantile (IQR inter-quartile range). **H** Expression level of VIM in organoids and tumoroids with or without macrophage co-culture. The height of the box ranges from the first quartile to the third quartile. The upper whisker indicates maxima but does not exceed 1.5 * IQR from third quartile. The lower whisker extends to the minima or 1.5 * IQR from the first quantile (IQR inter-quartile range). One-sided $t$ test was performed with the alternative hypothesis that VIM expression increased upon co-culturing with macrophage. **I** Recapitulating in vivo receptor-ligand interactions in organoid co-cultures, the graph edges represent interactions between macrophage and tumor cells. Black edges show upregulated R/L interactions in co-culture, with circles for receptors and triangles for ligands. **J** Proximity-ligation assay (PLA) performed in tumoroid-macrophage coculture ($n = 3$). Red dots indicate SPP1-CD44 interactions (Scale = 50 μm). In all panels, *$p \le 0.05$; **$p \le 0.01$; ***$p \le 0.001$; ****$p \le 0.0001$; ns: $p > 0.05$. Source data are provided as a Source Data file.

and in immune cell migration and inflammation, a reflection of the absence of TME in these cultures. While these features of organoid culture may prove limiting for some studies (e.g., immune therapies targeting a cell-surface receptor that may become suppressed in tumoroid culture), this reductionist system enables interrogation of the functional consequences of individual TME-carcinoma interactions.

To this end, we focused on the macrophage population as it is a major component of the TME and has been associated with both pro- and anti-tumorigenic activities. Our profiling of macrophages in treatment-naïve CRC (primarily MSS) was consistent with that of recent studies employing single cell transcriptomic analyses of colorectal cancers[15–20]. We observed three predominant macrophage states: an antigen-presenting C1QC+ state predominant in the normal colon, a pro-inflammatory IL1B+ state predominantly found in the normal liver, but also in normal colon and adenocarcinoma, and finally, a SPP1+ state enriched in tumors. Reassuringly, we identified these same states in three commonly employed mouse models of microsatellite-stable (MSS) CRC, with roughly similar distributions observed in human tumors, highlighting the value of immune-competent mouse models of human cancer.

*SPP1* encodes the extracellular matrix component Osteopontin, which acts as a CD44 ligand and has pleiotropic roles in tumor biology, most notably suppression of T cell activation[9], and the SPP1+ macrophage state is associated with immune evasion and metastatic proclivity[15,21,22]. Prior studies suggest that polarization of macrophages toward this SPP1+ state is a result of TME properties, including oxygen tension, the presence of FAP+ cancer-associated fibroblasts, and ECM composition, consistent with our pathway analysis of SPP1+ macrophages[22,25,40]. Recent single cell transcriptomic studies hypothesize a circulating monocyte origin for these SPP1 + TAMs[25]. Using the TME-free tumoroid cultures, we were able to directly test how CRC tumoroids influence macrophage polarization, revealing a strong induction of the SPP1+ state in human macrophages derived from circulating monocytes, indicating that carcinoma cells themselves can induce this state in the absence of cancer-associated fibroblasts (CAFs) or a hypoxic environment. We also found that CAFs alone were weak inducers of the SPP1+ macrophage state, but their addition to tumoroid-macrophage co-cultures potentiated polarization toward this immunosuppressive macrophage state. Conversely, the ability of SPP1+ macrophages to influence carcinoma cell identity appeared to correlate with expression levels of CD44, the Osteopontin receptor. In

tumoroids that were responsive to macrophage co-culture, induction of EMT transcriptional programs were observed, consistent with prior reports on pro-metastasis and EMT promoting functions of Osteopontin[42,44].

Ultimately, this study uncovers shortcomings and advantages of human patient-derived tumor organoid culture systems, providing a framework for utilizing these systems in colorectal cancer research and therapeutic development.

# Methods
## Human tissues
Human colorectal cancer specimens were obtained from patients undergoing elective surgery at Hospital of the University of Pennsylvania with written informed consent under the protocol approved by the University of Pennsylvania Institutional Review Board (Protocol number 827759 and PI - Dr. Bryson Katona). For those individuals who consent, after surgical resection, tumors are received in pathology, grossly examined by a pathologist, and a section of tumor and normal adjacent colon is obtained when extra tissue can be provided for research use without compromising patient care. Tissue is then transferred into ice cold phosphate-buffered saline (PBS) and placed on ice until use.

## Mouse models
All mouse-related protocols in this study were sanctioned and overseen by the University of Pennsylvania's Institutional Animal Care and Use Committee (IACUC) (Animal Welfare Assurance Reference: A3079-01; Approved protocol: 803415 attributed to Dr. Lengner). These procedures align with the National Institutes of Health's Guide for the Care and Use of Laboratory Animals. The *Apc*^min mice utilized were from The Jackson Laboratory (Strain #:002020). Our study incorporated both male and female mice, and all were kept within a C57BL/6 J genetic environment. Animals were euthanized before reaching any of the following criteria: tumor size reaches 1.0 cm diameter, bloody stool, weight loss (20% of initial body weight), or body condition score equal to or below 2 using the Ullman-Cullere MH Lab Animal Sci 1999;4 criteria. The maximal tumour size/burden never exceeded the threshold above.

## Tissue dissociation and cell sorting
Tumor and normal colon tissue was dissociated at 37 °C using a commercial Human Tissue Dissociation Kit (Miltenyi Order no. 130-095-929) following manufacturer's instruction. Briefly, 50–100 mg

tissue was minced into small pieces using a surgical blade and then incubated with 5 ml dissociation solution for 30 min on a tube rotator at 12 rpm. Tissue fragments settling down at the bottom of the tube are collected and minced again. DNase I (NEB Order no. M0303L) is then added to the dissociation solution to the final concentration of 10 U/ml (200X) and incubated at 37 °C for another 30 min. The cell pellet is resuspended in 1 ml Red Blood Cell Lysis Solution (Sigma Order no. 11814389001) for 4 min at RT and diluted to 15 ml using HBSS buffer containing 1% BSA. Any residual fragments are removed by filtering samples just prior to cell sorting through a 35-μm nylon mesh. All centrifugation steps are set to 250 x g for 5 min at 4 °C. DAPI (Thermo Fisher Order no. 62248) was used for living versus dead discrimination. Cells were sorted using a Becton Dickinson (BD) FACS Aria controlled by BD FACS DIVA software. FSC-H, FSC-W, SSC-H, SSC-W in combination serve to exclude doublets. Cells were sorted into an Eppendorf protein low-binding tube containing HBSS with 0.04% BSA.

## Single-cell RNA-seq
Sorted cells were immediately processed using a10x Genomics Chromium controller and the Chromium Single Cell 3' Reagent Kits V3 protocol. 7000–16,000 cells were loaded for each sample. Cells were partitioned into gel beads, lysed, and barcoded through reverse transcription. cDNA was purified and amplified using appropriate cycle number following the manufacturer's protocol. Libraries were constructed using 10x Genomics Library Prep Kit. Library quality was checked using Agilent High Sensitivity DNA Kit and Bioanalyzer 2100. Libraries were quantified using dsDNA High-Sensitivity (HS) Assay Kit (Invitrogen) on Qubit fluorometer and the qPCR-based KAPA quantification kit. Libraries were sequenced on an Illumina Nova-Seq 6000 with 28:8:0:87 paired-end format.

## Organoid culture
Organoid culture was carried out following the method previously described[38] from fresh surgically resected human colorectal adenocarcinomas or adjacent normal colon. For normal colonic crypt culture, the surface epithelial lining was first removed by surgical scalpel. The remaining tissue was washed vigorously with cold DPBS containing Pen/Strep and Gentamicin for 10 times before incubating with 2.5 mM EDTA (Invitrogen, AM9260G) for 15 min at 4 °C. Crypts were liberated by gently pushing the tissue through a 25 ml serological pipette against the bottom of the tube with a narrow gap in between. Crypts were suspended in Matrigel and dispensed onto 48-well-plate in a 100 μl drop. Tumor sample dissociation followed the protocol detailed in the cell sorting section above with the following modification: instead of enriching the single cell fraction, tissue clusters isolated in the cell strainer were collected and resuspended in Matrigel (BD, 356231) and seeded in a 100 μl drop. The final seeding density is 100 intact crypts per drop. Medium composition is as follows: Advanced Dulbecco's Modified Eagle's Medium/F12 (Thermo Fisher Scientific, 12634-010) was supplemented with penicillin/streptomycin (Thermo Fisher Scientific, 15140-122), 10 mM HEPES (Thermo Fisher Scientific, 15630-080), 2 mM GlutaMAX (Thermo Fisher Scientific, 35050-061), 1 X N2 Supplement (Thermo Fisher Scientific, 17502-048), 1 X B-27 Supplement (Thermo Fisher Scientific, 17504-044) to prepare a basal medium. An expansion medium was made by supplementing the basal medium with 10 nM gastrin I (Sigma-Aldrich, G9145-.1MG), 1 mM N-acetylcysteine (Sigma-Aldrich, A9165-5g), 100 ng/ml recombinant mouse Noggin (PeproTech, 250-38), 50 ng/ml recombinant mouse EGF (Thermo Fisher Scientific, PMG8041), 1 ug/ml recombinant human R-spondin1 (R&D, 4645-RS-025), 500 nM A83-01 (Tocris, 2939),10 uM SB 202190 (Sigma-Aldrich, S7067-5MG). The organoids were passaged biweekly by digesting with TyrpLE (Thermo Fisher Scientific, 12604013) for 5 min at 37 °C and then splitting in 1:3-5 ratio. The medium was supplemented with 10 uM Y-27632 (Selleck, S1049) for the first two days of culture after seeding.

## Monocyte isolation and macrophage differentiation
Normal donor human monocytes were procured from the Human Immunology Core at the University of Pennsylvania. Cells were cultured in RPMI-1640 medium (with 10% FBS, 1% Pen/Strep and 2 mM glutamine). Cells were treated with 50 ng/ml human M-CSF (PEPROTECH, Cat# 300-25-50ug) for 7 days. Then collect cells. Macrophage cell marker, CD16 (BD biosciences, Cat# 335806), was detected by flow cytometry for verification purpose.

## Organoid-Macrophage-CAF co-culture
Two days before the co-culture procedure, organoids were dissociated into single cells and seeded within Matrigel drops in 12-well-plates. When setting up the co-culture, organoids were released from the Matrigel by incubating with Corning Cell Recovery Solution for 20 min at 4 °C. Intact organoids were mixed with $10^5$ monocyte-derived macrophages (or macrophages alone) and seeded back to the 12-well-plate. The culture was maintained for 48 h and then harvested for analysis. For Macrophage conditioned media experiments, a total of 1000 cells were placed in a Matrigel matrix (100 μl) and allowed to proliferate for a duration of four days. Two days prior to collecting the conditioned medium, 600 μl of fresh medium were introduced. At the time of harvest, 600 μl of medium were removed and filtered through a 0.22 um syringe filter. Subsequently, 500 μl of filtered medium were introduced into the macrophage cultures, maintaining a 1:1 ratio with fresh medium. CRC-associated fibroblasts (CAFs) were generously provided by Dr. Kathryn Hamilton's lab at Children's Hospital of Philadelphia. CAFs were isolated from a Stage IIA colorectal cancer patient. CAFs were seeded on a tissue culture plate pre-coated with bovine type I collagen (Advanced Biomatrix 50360232-5010-D, 1:100 dilution in PBS), and established and expanded in Advanced DMEM:F12 culture medium supplemented with 15% FBS (HyClone) and Primocin (Invivogen, 1:500 dilution). After passage two, the cells were maintained in DMEM containing 15% FBS (HyClone) and 1% penicillin/streptomycin. For each co-culture experiment involving CAFs, $10^4$ fibroblasts were seeded with either organoids, macrophages, or both.

## Proximity ligation assay (PLA)
The PLA protocol was performed in accordance with the manufacturer's instructions (Duolink In Situ PLA Kit, Sigma-Aldrich). To begin, epithelial cell-macrophage coculture was established in a Matrigel drop. At the end of the culture period, the samples were fixed in 2% paraformaldehyde (PFA) at 4 °C for 20 min with gentle shaking. The organoid pellet was recovered by centrifugation at 400 xg for 5 min and embedded in low-melting temperature agarose drop. Staining was performed on slides. The slides were incubated with blocking buffer in a heated humid chamber for 60 min at 37 °C, then washed incubated overnight at 4 °C with primary antibodies against CD44 [Novus Biologicals, NBP1-47386, 1:400] and SPP1 [Rockland, 600-401-ET6,1:200]. After washing, the slides were incubated with PLA probes for 1 h at 37 °C. The PLA probes consisted of corresponding oligonucleotide-conjugated secondary antibodies against the primary antibodies. Subsequently, the slides were washed with wash buffer, and incubated with ligation solution for 30 min at 37 °C to ligate the PLA probes in proximity. Following this, the slides were washed with wash buffer and incubated with amplification solution for 100 min at 37 °C to amplify the ligation products. The images were acquired using a Leica DMi8 Inverted LED Fluorescence Phase Contrast Microscope. Control (Ctl) PLA is conducted in the absence of SPP1 antibody.

## Single molecule fluorescence in situ hybridization (smFISH)
The co-detection of human *SPP1* RNA scope probes and CD68 was performed using RNAscope® Multiplex Fluorescent v2 Assay (Cat# 323100) and Immunofluorescence - Integrated Co-Detection kit (Advanced Cell Diagnostics) according to the manufacturer's instructions. Briefly, tissue sections were deparaffinized (2 × 5 min fresh

xylene, 2 × 2 min 100% ethanol at room temperature) and completely dried in oven for 5 min at 60 °C. Hydrogen peroxide was then applied for 10 min at room temperature. After being washed twice with distilled water, the slides were slowly submerged into hot 1x co-detection target retrieval solution for 15 min (98–102 °C). Slides were then immediately transferred into distilled water, washed twice, then once with 1x PBST (0.1% Tween-20) and then incubated with primary antibody overnight at 4 °C. The primary antibody diluted in co-detection antibody diluent: E-Cadherin (24E10) Rabbit mAb (1:50, CST), anti-CD68 (1:100, Abcam). Following primary antibody incubation and washing 3 × 2 min with PBST, post-primary fixation occurred by submergence in 10% Neutral Buffered Formalin (NBF, VWR) for 30 min at room temperature. Protease plus was then applied for 30 min at 40 °C in a HybEZ™ Oven and slides were then washed with distilled water 2 × 2 min. Next, 1:50 diluted SPP1 (Cat# 420101-C2), positive control (Cat# 320881) and negative control (Cat# 320871) probes were applied followed by hybridization for 2 h at 40 °C in a HybEZ™ Oven. After probe hybridization, the slides were hybridized with AMP1 for 30 min, AMP2 for 30 min, AMP3 for 15 min, 1:750 diluted Opal™ 570 (Cat# FP1488001KT, Akoya Biosciences) fluorophore in RNAscope® Multiplex TSA buffer for 30 min, HRP blocker for 15 min, HRP-C2 signal for 15 min (for SPP1 probe), 1:750 diluted Opal™ 690 (Cat# FP1497001KT, Akoya Biosciences) fluorophore in RNAscope® Multiplex TSA buffer for 30 min, HRP blocker for 15 min in a HybEZ™ Oven at 40 °C, successively. The slides were washed in 1x wash buffer twice for 2 min at room temperature between hybridization steps. After the final wash step, fluorophore- conjugated secondary antibody was applied (diluted as 1:500 in co-detection antibody diluent) for 1 h at room temperature and followed by washing with 1x PBST 3 × 2 min at room temperature. The slides were then counterstained with DAPI for 30 seconds without washing and mounted using Prolong Gold antifade mounting medium (Thermo Fisher Scientific) for imaging.

### Single cell RNAseq data processing

Sequencing reads for the human and mouse samples were first preprocessed with 10x Genomics Cell Ranger pipeline and aligned to the GRCh38 reference and GRCm38 (mm10) reference genome respectively. An initial filtering was performed on the raw gene-barcode matrix output by the Cell Ranger cellranger count function, removing barcodes that have less than 1000 transcripts (quantified by unique molecular identifier (UMI)) and 500 expressed genes ("expressed" means that there is at least 1 transcript from the gene in the cell). Barcodes that pass this filter were considered as cells and were included in the final dataset. Furthermore, cells with high levels of mitochondrial genes (greater than 20% of total UMI count) were excluded for all downstream analysis. For samples multiplexed using the TotalSeq-B protocol, cells were demultiplexed by performing Louvain clustering on the UMAP generated with the hashtag count matrix. Gene-barcode UMI count matrix combined from all datasets was size-factor corrected and log transformed to produce a normalized gene expression matrix.

### Dimension reduction, clustering and cell type annotation

We used the VisCello package to generate a series of PCAs and UMAPs for different cell subsets. The processing pipeline was described previously in Zhu et al.[39]. Briefly, we applied an "informative feature (IFF) selection" procedure to select genes that have high gini coefficient which indicates the "inequality" (therefore specificity) of the gene's expression across clusters. Principal component analysis (PCA) was then performed on the IFF-cell matrix, and the top PCs were used as features for the UMAP algorithm. UMAP was computed using the *umap* function in the uwot R package, with "cosine" distance metric, 30 nearest neighbors, and the rest of the parameters as default. Louvain clustering was run on the k-nearest neighbor graph (k = 20) constructed from cell embeddings on the UMAP. We annotated each of the

clusters based on comparing cluster-specific differentially expressed genes with known cell-type marker genes based on previous literature. Top DEGs/cell-type markers include: Epithelial: *EPCAM*, Fibroblast: *VIM,COL1A1,COL1A2*; Myofibroblast: *VIM*, *ACTA2*; Endothelial: *CDH5,CLDN5,ESAM*; T cells: *CD3D,CD3E*; B cells: *CD19,CD20*; Plasma cells: Immunoglobulin genes, such as *JCHAIN, IGHA, IGHG, IGHM*; Macrophage: *CD68,LYZ*; DC: *CD1C,BATF3*; Mast cells: *TPSAB1, TPSD1, CPA3*. A small cell population from liver/liver metastasis samples form a separate cluster and were broadly annotated as "liver cells" as we do not have enough resolution to distinguish the cell subtypes.

### Differential expression analysis

Differential expression analysis was carried out using the "sSeq" algorithm, with FDR < 0.05 and log2 fold change > 1 as cutoff for differentially expressed genes (DEGs). For DE between liver metastasis and primary tumor shown in Supplemental Fig. 2, Mann-Whitney *U* test was used instead because of the limited cell number and sequencing depth.

### Pathway/signature enrichment analysis

To compute a per-cell enrichment score for each pathway in the Reactome database or cell-type specific signature shown in Figs. 3D, S3E, S4H, 5E, F, S5, 6C, G and S7B, we utilized the AUCell package. The AUCell package uses the "Area Under the Curve" (AUC) to calculate the enrichment of a particular gene set among top expressed genes of each cell. Quantification of pathway activity and cell-type signature score enables subsequent statistical test, such as ANOVA and Tukey's test in Fig. 6C, and Student's *t* test to derive up-regulated Reactome pathways in liver metastasis shown in Supplemental Fig. 3E.

Gene ontology (GO) enrichment analyses in Figs. 3F, 4E, S4D, S6A, B were computed using the ClusterProfiler package with *q*-value cutoff of 0.05 and ontology type "Biological Process" (BP).

### Carcinoma-TME interaction analysis

Receptor/ligand information was obtained from a curated database[40]. To predict receptor-ligand interactions that are specifically induced by the tumor microenvironment, we took advantage of the normal adjacent epithelial cells we collected and used these as the control to derive receptor/ligands that are specifically up-regulated in the carcinoma cells. For each of the receptors/ligands significantly up-regulated (FDR < 0.05), we looked for its counterpart in the non-epithelial TME cells. TME cell types that differentially express the receptor/ligand counterpart were considered to be communicating with the tumor epithelial cell via the receptor-ligand interaction. The receptor-ligand interactions between tumor epithelial cell and TME cells were visualized as a bipartite graph using the ggraph package.

### Macrophage state identification and and FateID analysis

Monocytes/macrophages were initially identified through the expression of classic marker genes such as *CD14*, *CD68*, and *LYZ*. Subsequently, Louvain clustering was applied to the subset of cells, revealing substantial heterogeneity within the monocyte/macrophage cell population. To categorize the various cell states within this population, a one-vs-rest differential gene expression analysis was conducted, resulting in lists of genes that exhibited differential expression. The top differentially expressed genes were then compared with well-established macrophage state marker genes from previous literature sources[15,18]. Commonly recognized markers were employed to annotate the cell states. Three key state marker genes, namely *IL1B*, *SPP1*, and *C1QC*, were utilized to segregate the cell populations into three distinct groups, aligning with the results obtained through Louvain clustering. To probabilistically assign cells to each of these states, state-specific genes were redefined through differential expression analysis. Subsequently, FateID analysis was performed using these signature genes and default parameters. Finally, pathway enrichment analysis was conducted on cells falling within these three states.

## Consensus molecular subtyping of colorectal cancer

Consensus molecular subtypes were called using the CMScaller R package[8]. For each sample, we aggregated its single-cell gene expression counts to obtain a pseudo-bulk expression vector. We run CMScaller with parameter *RNASeq* = TRUE and the rest parameters as default. CMScaller performs classification using nearest template prediction algorithm with pre-defined cancer-cell intrinsic CMS templates (Supplementary Data 4).

## CODEX antibody conjugation

Akoya antibodies were purchased preconjugated to their respective CODEX barcode (Supplementary Data 5). All other antibodies were custom conjugated to their respective CODEX barcode (Supplementary Data 5) according to Akoya's CODEX user manual using the CODEX Conjugation Kit (Akoya, 7000009). Briefly, 50 µg of carrier-free antibodies (Supplementary Data 5) were concentrated by centrifugation in 50 kDa MWCO filters (EMD Millipore, UFC505096) and incubated in the antibody disulfide reduction master mix for 30 min. After buffer exchange of the antibodies to conjugation solution by centrifugation, addition of conjugation solution and centrifugation, respective CODEX barcodes resuspended in conjugation solution were added to the concentrated antibody and incubated for 2 h at room temperature. Conjugated antibodies were purified by 3 buffer exchanges with purification solution. 100 µl of antibody storage buffer was added to the concentrated purified antibodies.

## CODEX staining

CODEX staining was done using the CODEX staining kit (Akoya, Cat 7000008) according to Akoya's CODEX user manual with modifications to include a photobleaching step and overnight incubation in antibodies at 4 °C. FFPE samples were sectioned at 5 µm and mounted onto 22 mm × 22 mm coverslips coated with poly-L-lysine (Sigma-Aldrich, P8920) coated according to Akoya's CODEX user manual. Sample coverslips were heated on a 55 °C hot plate for 25 min to bake the tissue. Sample coverslips were deparaffinized in xylene and rehydrated in a graded series of ethanol (2 times 100%, 90%, 70%, 50%, 30% and 2 times ddH2O). Antigen retrieval was performed in 1X Tris-EDTA buffer pH 9.0 (Abcam, ab93684) with a pressure cooker for 20 min. After equilibrating to room temperature, sample coverslips were washed 2 times with ddH2O and submerged in a 6-well plate containing 4.5% H2O2 and 20 mM NaOH in PBS (bleaching solution) for photobleaching. The 6-well plates were sandwiched between two broad-spectrum LED light sources for 45 min at 4 °C. Sample coverslips are transferred to a new 6-well plate with fresh bleaching solution and photobleached for another 45 min at 4 °C. Sample coverslips were washed 3 times in PBS and then 2 times in hydration buffer. Sample coverslips were equilibrated in staining buffer for 30 min and incubated in the antibodies (Supplementary Data 5) diluted in staining buffer plus N Blocker, G Blocker, J Blocker, and S Blocker overnight at 4 °C. After antibody incubation, sample coverslips were washed 2 times in staining buffer and fixed for 10 min in 1.6% paraformaldehyde (Electron Microscopy Sciences, 15710) in storage buffer. Sample coverslips were washed 3 times in PBS and incubated in ice cold methanol for 5 min. After incubation in methanol, sample coverslips were washed 3 times in PBS and incubated in final fixative solution for 20 min. The sample coverslips were then washed 3 times in PBS and stored in storage buffer.

## CODEX imaging

CODEX reporters were prepared according to Akoya's CODEX user manual and added to a 96-well plate. The CODEX instrument was set up for a CODEX run according to Akoya's CODEX user manual using the CODEX instrument manager software. Details on the order of fluorescent CODEX Barcodes can be found on Supplementary Data 5.

Images were taken with a Nikon Plan Apo λ 20X/0.75 objective on a Keyence BZ-X810 fluorescence microscope. Microscope setup was done according to Akoya's CODEX user manual with a z plane of 11 and z pitch of 1.2 µm. 2 areas of 1.5 mm × 1.1 mm were imaged for patient 86 and patient 92. 3 areas of 2.0 mm × 1.5 mm were imaged for patient 28.

## CODEX image analysis

The CODEX images were processed with the CODEX® processor software, which performs background subtraction, deconvolution, stitching and segmentation with default settings. The segmented data were imported into R, where a two-component gaussian mixture model was fit to the intensity values of each channel. The component with a lower mean was treated as background noise and only signals from the higher component was retained. The corrected intensity values were then log transformed and imported into VisCello[47,48] for clustering, cell type annotation and visualization.

## Reporting summary

Further information on research design is available in the Nature Portfolio Reporting Summary linked to this article.

## Data availability

All raw sequencing data and processed data generated and analyzed in this study have been deposited in the Gene Expression Omnibus (GEO) database of the National Institutes of Health (NIH) under accession number [GSE203608]. This ensures that the data is publicly available and can be accessed by other researchers to replicate and verify the results reported in this study. Human single cell data that is compatible with VisCello has also been deposited to Zenodo[46] (https://zenodo.org/record/7872684). Readers can utilize VisCello (https://github.com/qinzhu/VisCello) to interactively explore and analyze the deposited data, which include: an R data object (eset.rds) containing the raw count matrix and normalized expression matrix, as well as meta data for all cells; the clist.rds file containing a list of dimension reduction results for different subsets of the data. The mouse transcriptome sequencing datasets generated is available in the NCBI Gene Expression Omnibus (GEO) with GSE198759. Source data are provided with this paper.

## Code availability

The code related to the analysis has been deposited to Gibhub. [https://github.com/qinzhu/ColonCancerManuscript_CodeRepo].

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

## Acknowledgements

This work was funded by a Translational Center for Excellence award from the Abramson Cancer Center at the University of Pennsylvania (P30CA016520). C.J.L. is supported by R01 CA168654. K.T. is supported by U01 CA243072 and U2C CA233285. N.L. is supported by R50 CA221841. B.W.K. is supported by Scott and Suzi Lustgarten Colon Cancer Research Fund. A.T.-T. is supported by R01 CA196299. M.H. is supported by R37 CA234027, P01 CA265794, and CRI Lloyd J. STAR grant. A.K.R. and C.J.L. are supported by R01 CA272903 and A.K.R. is supported further by P30CA013696. This work was supported by core facilities funded by the NIDDK P30 Center for Molecular Studies in Digestive and Liver Diseases (P30DK050306) at the University of Pennsylvania and P30CA013696 at Columbia

University. Cartoon models were created with BioRender.com with full license to publish.

## Author contributions

N.L., Q.Z., K.T. and C.J.L. conducted and designed most of the experiments, with exceptions detailed below. Y.T. and X.W. assisted with specimen processing and organoid banking. P.S. and A.T-T. carried out the RNA-Seq experiment and subsequent data interpretation. K.J.A. completed the CODEX experiment. P.Y., I.W.F. and M.H. provided support for Macrophage related experiments. J.J. oversaw patient consent procedures and specimen gathering. N.N.M., C.B.A. and R.E.R. supplied clinical specimens. E.E.F. provided pathological diagnoses. Z.C. and S.A-T. offered general lab support. B.W.K. and A.R. held the clinical specimen collection protocol from the University of Pennsylvania. N.L., Q.Z., K.T., B.Z.S. and C.J.L. contributed to data interpretation, presentation and manuscript preparation.

## Competing interests

The authors declare no competing interests.

## Additional information

[1]Department of Biomedical Sciences, School of Veterinary Medicine, University of Pennsylvania, Philadelphia, PA 19104, USA. [2]Institute for Regenerative Medicine, University of Pennsylvania, Philadelphia, PA 19104, USA. [3]Department of Pediatrics, Perelman School of Medicine, University of Pennsylvania, Philadelphia, PA 19104, USA. [4]Division of Oncology and Center for Childhood Cancer Research, Children's Hospital of Philadelphia, Philadelphia, PA 19104, USA. [5]Department of Medicine, Perelman School of Medicine, University of Pennsylvania, Philadelphia, PA 19104, USA. [6]Department of Pathology and Laboratory Medicine, Perelman School of Medicine, University of Pennsylvania, Philadelphia, PA 19104, USA. [7]Division of Cancer Pathobiology, Children's Hospital of Philadelphia, Philadelphia, PA 19104, USA. [8]Division of Colorectal Surgery, Department of Surgery, University of Pennsylvania, Philadelphia, PA 19104, USA. [9]Division of Endocrine and Oncologic Surgery, Department of Surgery, University of Pennsylvania, Philadelphia, PA 19104, USA. [10]Abramson Cancer Center, Perelman School of Medicine, University of Pennsylvania, Philadelphia, PA 19104, USA. [11]Division of Digestive and Liver Diseases, Department of Medicine, Herbert Irving Comprehensive Cancer Center, Vagelos College of Physicians and Surgeons, Columbia University Irving Medical Center, New York City, NY 10032, USA. [12]These authors contributed equally: Ning Li, Qin Zhu. ✉e-mail: Ningli@upenn.edu; Tank1@chop.edu; Lengner@upenn.edu

