## [Peer Review File · Nature Communications]

REVIEWER COMMENTS

Reviewer #1 (Remarks to the Author): Expert in colorectal cancer organoids and mouse models, and functional genomics

The manuscript submitted by Li et al., entitled "Mapping and modelling human colorectal carcinoma interactions with the TME" uses single-cell RNAseq methodology to describe in detail how cancer cells interact with their environment. To this end, they analyze data from human colorectal cancer samples, liver metastases, mouse models samples and 3D colorectal cancer organoids derived from these primary tissues.

The manuscript is very descriptive however the data provided if published would increase the knowledge about how cancer cells interact with the environment, especially immune populations.

Main comments:

1. The manuscript does not take into account the extracellular matrix as a main component of the tumour microenvironment. The title and abstract describe the work as how cancer cells interact with the TME when it is not stated that has the big limitation that ECM is not being covered. It has been shown by Sato's group
2. There is a lack of novelty on the manuscript data. If this work would have been submitted 2-3 years ago, then the data would be novel but recent manuscripts provide similar information (Chen et al. 2021, Liu et al. 2022, Cañellas-Socias et al, 2022.). Authors should clarify what is the novelty and authors should work on the data obtained to provide more novelty to this study. Figure 1 is just informative. Figure 2 and Figure 3 are findings that have been described before and expected. Data with mouse samples covers similar findings and the type of immune-cancer cell interactions has been also suggested before. Only Figure 6 that shows co-culture experiments provides a bit more of functional data. The discussion should also try to highlight the novelty of the manuscript.
3. To contribute to cancer research, the scRNAseq data would need to be uploaded in a platform that can all users have access (e.g. single cell portal from broadinstitute).
4. Liver metastases results are unclear. I understand the limitations to work with liver metastases but there are only 4 samples, most of the sample is non-epithelial and number of cells analysed are less than 6,000. Most of the findings are described in supplementary but I consider that may not be significant. I suggest authors to think about how significant these results are.

5. The findings shown in the figures are vague described in the text. It many times says " most of the pathways..", "many suppressed..." "found a striking suppression of gene expression programs.." but does not specify numbers, %, significance, statistics, or any details about the findings.

6. There can be a different TME depending on the stage of tumour, can this be identified by the scRNAseq analysis? or is it needed a higher number of samples?

Minor comments:

1. Figure 1c, it should be written colonoid and tumoroid as has been done in the text.

2. Figure 1b, tumours could be also ordered following stages to clarify the type of samples, and figure 1d can also be ordered from left to right by stages. Can the authors include images of all CRC organoids? in figure 1d, the organoids phenotype can be hardly seen.

3. It is written "for at least 4 passages", but nothing about maximum of passages done in these two months.

4. Figure 2c, it cannot be seen the stainings of each marker as the magnification is low and the individual stainings are not shown. This happens with some other stainings throughout the text.

5. This sentence should be re-written: "To this end, we asked how well human tumoroids or various mouse models (including the Apcmin model of familial adenomatous polyposis, the AOM-DSS model of inflammation-driven colorectal cancer, and the APKS model of invasive, metastatic colorectal adenocarcinoma generated by endoscope-guided orthotopic implantation of

CRISPR/Cas9-engineered tumoroids with Apc, p53, Kras, and Smad4 mutations into the colonic mucosa of syngeneic mice) recapitulate human primary tumors and tumoroid model tumoroids or various models"

It is long and not clear for the reader.

Reviewer #2 (Remarks to the Author): Expert in single-cell and spatial omics, colorectal cancer and tumour immune microenvironment

Colorectal cancer (CRC) is one of the most common malignant tumors worldwide and the global burden and lack of major therapeutic advances in CRC give rise to high mortality rates. Mouse tumor model plays a very important role in the research of colorectal cancer, but invasive adenocarcinoma and gene editing will result in the premature mortality of the model. Recently, the development of human organoids enables the modeling of a range of immune checkpoint blockades, but the co-culture of organoids has difficulty reproducing the interaction between tumor cells and the tumor microenvironment.

In this manuscript, the authors collected 17 CRC patients and conducted single-cell RNA sequencing with the primary tumor, normal adjacent tissues, metastatic lesions. Also, the authors conducted single cell RNA sequencing with normal organoids and tumoroids cultured from human epithelium, which is very valuable data. The authors compared the advantages and limitations between mouse model and organoids in mapping and modeling human colorectal carcinoma interactions with the tumor microenvironment, which will be helpful for selecting efficient methods in oncology research. However, it has been reported that SPP1+ Macrophage is highly related to the progression of tumor, the innovation is insufficient in biological meaning. Also, the authors need to address the following concerns:

Major comments:

1. In Figure 3B and 3D, primary normal adjacent epitheliums are more similar to mature absorptive colonocytes and primary tumor are more similar to crypt base columnar stem cells. Human normal and tumor epitheliums have significant difference in transcriptional identities, while the normal organoids and tumoroids both shift towards stem cell. Have the organoids lost the characteristics of human normal and tumor tissues during the in vitro culture? The authors should show more evidence to validate the correlation between organoids and the human epithelial samples.
2. In addition to macrophages, the T and B cells are also play important roles in the epithelium-immune interaction. In Figure 5A, the authors tried to model the human CRC TME in mouse model, but there are few T and B cells. More mouse samples may be helpful to increase the proportion of T and B cells. And the authors need to supplement the detail and metadata of mouse model scRNAseq to the methods.
3. In Figure 6C, the authors claimed that there is a clear shift in macrophage polarization toward the SPP1+ state. Whether this phenomenon can be observed in P08 and P24? And the normal organoids and tumoroids seem to have a similar influence on macrophages during the co-culture.

Minor comments:

1. In Figures 2A and 5A, the authors need to check the small populations of cells far away from the other cluster, and filter them if they are contaminated.

2. In Figure 2D, the y-axis may not be the cell type. As the author described, it may be the patient's ID.

Reviewer #3 (Remarks to the Author): Expert in colorectal cancer clinical and molecular research, and tumour-associated macrophages

Tumor-associated macrophages (TAMs) constitute the dominant immune cell population in colorectal cancer (CRC). Recent studies have shown that some of these cells are continuously recruited from the blood and, upon interaction with factors released in the tumor microenvironment (TME), express specific genes, such as SPP1, and become immunosuppressive, thus contributing to generate a niche that sustains cancer cell growth and diffusion. By using single-cell transcriptomic analyses of treatment-naïve, surgically resected human CRC samples, Li and colleagues sought to map putative interactions between human CRC cells and TAMs. In parallel, by establishing CRC organoids (tumoroids) and normal colon organoids, the authors assessed the effect of cancer cells on TAMs differentiation. As expected, primary tumors contained a variety of cell types, including immune and non-immune cells. Next, by using CODEX analyses, the authors showed that carcinoma cells had a higher propensity for interaction with stromal cells and immune cells relative to their normal counterparts, and documented potential receptor-ligand interactions between carcinoma cells and cells within their microenvironment. It was also shown that the average transcriptome of cells from tumoroids and organoids was quite similar while there were differences between primary tumors and the derived tumoroids. In this context, several genes were suppressed and others were activated in tumoroids as compared to primary carcinoma cells. Specifically, genes involved in communication with immune cells were suppressed while those related to cell division, patterning, and metabolism were generally activated in tumoroids consistent with the shift to stem and progenitor cell states and away from terminally differentiated absorptive states in culture. The latter findings were then seen in various mouse models of colon tumorigenesis. In the last parts of the study, the authors focused on the distribution of normal and tumor macrophages and showed that macrophages in the SPP1+ state, which is known to suppress anti-tumor immune responses, are more prevalent in tumors relative to the other states. Consistently, stratification of colorectal cancer TCGA transcriptomes by SPP1 signature enrichment revealed poor survival in patients exhibiting high SPP1 signature. SPP1 itself encodes osteopontin, a secreted ECM component, and ligand for CD44, the latter being up-regulated in tumors relative to normal tissue. Similar findings were seen in all the mouse tumor models. Finally, culturing circulating monocyte-derived macrophages with tumoroids induced a shift in macrophage polarization towards the SPP1+ state.

Although the topic is of interest and this reviewer acknowledges the enormous amount of work performed by the authors to analyze the single-cell transcriptomic data, the manuscript is largely descriptive and several issues/points were not adequately taken into consideration by the authors. Specifically, the main conclusion of the study is a little bit skewed to convince the reader that differentiation of suppressive macrophages is exclusively driven by cancer cells.

Specific points

1. The elevated expression of SPP1 has been previously identified among TAMs across several cancers including primary CRC (Cell 2021;184(3):792-809 e23). The innovativeness of the manuscript is also partly hampered by a recent study showing that metastatic CRC express predominantly SPP1+ macrophages and these communicate with fibroblasts with the downstream effect of promoting immunosuppression in the TME (Clin Cancer Res. 2022 Oct 14;CCR-22-2041).
2. A major finding of the study is that cancer cells promote the differentiation of suppressive macrophages, which sustain tumorigenesis. No attempt was made by the authors to examine whether cancer cells either release factors that induce SPP1+ macrophages or such a differentiation relies on cell-cell interactions.
3. Although the authors documented potential receptor-ligand interactions between carcinoma cells and cells within their microenvironment, none of such interactions was eventually validated.
4. The in vitro studies do not help understand if the SPP1 phenotype induced by culturing tumoroids with circulating monocyte-derived macrophages is stable over time. Moreover, no mechanistic study was performed to assess the suppressive functions of the in vitro differentiated SPP1+macrophages.
5. Accumulating evidence indicates that differentiation and function of TAMs are mainly controlled by non-immune cells present in the TME (i.e. fibroblasts). Therefore, the in vitro studies should be implemented with further research aimed at evaluating the effect of cancer-associated fibroblasts (CAFs) on the differentiation of SPP1+ macrophages. Additionally, the authors should better characterize the different stromal cells present in the CRC microenvironment as well as the ability of SPP1+macrophages and CAFs to influence the tumor-associated T cells.
6. Do normal colon organoids promote differentiation of C1QC+ monocyte-derived macrophages? Is such a differentiation abolished/shifted by CAFs?
7. CRCs with high-level microsatellite instability (MSI) are marked by increased infiltration of T cells in the TEM. In contrast, CRCs with microsatellite stable (MMS) generally do not have significant levels of T cell infiltration and have a profoundly immunosuppressed TME (Hirano et al. 2021 Jap J Clinical Oncology, 51: 1, p. 10–19; Immunity 2016;44:698–711). The authors should assess whether the prevalent accumulation of SPP1+macrophages is restricted to CRC with MMS.
8. The authors stated that they were able to capture a total of 38,063 cells from the 15 primary tumors. Figure 1E shows the enormous variability among the tumor samples, and the number of cancer cells recovered from more than fifty percent of samples was very low (less than 2000) or absent. These findings, together with the lack of information regarding the procedure adopted to eliminate cells expressing genes indicative of cell death, introduce bias in the interpretation of the results.
9. Markers identified from the scRNA-seq results should be confirmed by protein analysis (i.e. immunostaining).
10. I could not find the suppl Tables indicated in the text.

REVIEWER COMMENTS

Reviewer #1 (Remarks to the Author): Expert in colorectal cancer organoids and mouse models, and functional genomics

The manuscript submitted by Li et al., entitled "Mapping and modelling human colorectal carcinoma interactions with the TME" uses single-cell RNAseq methodology to describe in detail how cancer cells interact with their environment. To this end, they analyze data from human colorectal cancer samples, liver metastases, mouse models samples and 3D colorectal cancer organoids derived from these primary tissues.

The manuscript is very descriptive however the data provided if published would increase the knowledge about how cancer cells interact with the environment, especially immune populations.

Main comments:

1. The manuscript does not take into account the extracellular matrix as a main component of the tumour microenvironment. The title and abstract describe the work as how cancer cells interact with the TME when it is not stated that has the big limitation that ECM is not being covered. It has been shown by Sato's group

Thank you for this comment, indeed the ECM is a critical component of the TME, however given that our study is based primarily upon single cell transcriptomics and cell-cell interactions in organoid cultures, there are relatively few insights that we can gain into ECM interactions from the datasets we have generated. We have altered the text to highlight this limitation.

Further, we now include a panel inferring carcinoma receptor-ECM interactions from the scRNAseq data, presented in Fig. S6C, D, and have also incorporated new data obtained from co-culturing cancer-associated fibroblasts, another important TME component, with tumoroids (in the presence or absence of macrophages) to expand upon the existing study (Fig. S10).

We asked if there are pathways associated with ECM interaction in the epithelial compartment. In the tumor vs. tumoroid (Fig. 3F) comparison we previously did, we found gene ontology related to "extracellular matrix organization" significantly enriched in the downregulated genes in tumoroid carcinoma cells relative to carcinoma cells *in vivo* (adjusted p-value = 3.5e-08, Supplemental Table 3). Interestingly, the reverse trend is seen in the normal colon vs. organoid comparison (Supplemental Table 3). In addition, we performed differential expression and gene ontology (GO) enrichment analysis between tumor vs. colon, and tumoroids vs. organoids. GO enrichment analysis indicates that genes associated with extracellular matrix organization are highly upregulated in carcinoma cells compared to normal epithelial cells (Fig. S6A, Supplemental Table 3). However, there are few differentially expressed genes (DEGs) related to ECM remodeling when comparing tumoroids vs. organoids (Supplemental Table 2).

To understand the difference in ECM-cell interaction between carcinoma and normal epithelial cells, and between *in vivo* and *in vitro* settings, we performed network analysis based on a curated ECM network database from matrixDB (Clerc *et al.* 2019 Nucleic Acids Res. 47:D376-D381.

doi:10.1093/nar/gky1035.), obtained from the *matrinetR* package (<https://rdr.io/github/KontioJuho/matrinetR/>). We computed pairwise gene expression correlations among carcinoma cells *in vivo* or from tumoroids and plotted the correlation as edge color on the ECM network (Fig. S6C, D). The overall correlations between the ECM genes are higher in tumors *in vivo* compared to tumoroids (Fig. S6E). The nodes of the networks are colored based on the log₂ fold change of DEGs between tumor and colon, and between tumoroid and organoid. Consistent with the GO analysis, the differences in multiple genes in the ECM network, including collagens (*COL1A1*, *COL1A2*, *COL3A1*, *COL4A1*), fibronectin (*FN1*), lumican (*LUM*) and osteonectin (*SPARC*), are highly significant *in vivo* compared to *in vitro*.

To understand the basis for these putative carcinoma-ECM interactions, we examine the mesenchymal component of the TME. Fibroblasts and myofibroblasts (an activated form of fibroblast, expressing several cytoskeletal genes such as *ACTA2*, Fig. S6F) are the major producers of ECM, and are particularly enriched in tumor samples (Fig. 2B, Fig. S6G). We found several genes up-regulated in cancer associated fibroblasts compared to normal fibroblasts. These include *CTHRC1*, *INHBA*, *BGN*, and *PDPN*, all of which are known to promote tumor progression (PMID32848510, 34346300, 29761248, 24354864).

2. There is a lack of novelty on the manuscript data. If this work would have been submitted 2-3 years ago, then the data would be novel but recent manuscripts provide similar information (Chen et al. 2021, Liu et al. 2022, Cañellas-Socias et al, 2022.). Authors should clarify what is the novelty and authors should work on the data obtained to provide more novelty to this study. Figure 1 is just informative. Figure 2 and Figure 3 are findings that have been described before and expected. Data with mouse samples covers similar findings and the type of immune-cancer cell interactions has been also suggested before. Only Figure 6 that shows co-culture experiments provides a bit more of functional data. The discussion should also try to highlight the novelty of the manuscript.

Indeed, ours is not the first study to interrogate the TME of colon cancer using single cell transcriptomics, and this is reflected by the fact that the prior studies in the past 2 years or so were published in *Cell*, *Nature*, and *Cancer Cell*, while our current manuscript is being considered at *Nature Communications*. Ultimately, however, the novelty of our study lies primarily in the development of patient-matched tumor organoid cultures, scRNAseq maps thereof, and their comparison to scRNAseq maps from primary tumors. To the best of our knowledge, this is the first study that addresses how long-term organoid culture alters the transcriptomic state of primary human carcinomas, and how these human models compare with state-of-the-art murine colon cancer models.

In the revised manuscript, we include further data analysis as suggested. Specifically, our bioinformatics analysis revealed a strong association between the SPP1+ macrophage state and the composition of immune components within the TME particularly the adaptive immune components (T and B cell subtypes, Fig. S8, S9), as well as with tumor stage and type. These analyses further support the model that SPP1+ macrophages exist in an immunosuppressive TME. Further, through newly included co-culture experiments with macrophages and cancer-associated fibroblasts derived from human CRCs (CAFs, Fig. S10), we demonstrate that the immunosuppressive macrophage state can be induced by carcinoma cells alone, and this is enhanced by the presence of CAFs. We find that CAF presence alone was a weak driver of

immunosuppressive macrophage polarization. Additionally, the conditioned media of carcinoma organoid cultures was insufficient to induce immunosuppressive macrophage polarization. These findings suggest potential mechanisms of immunosuppression and further illuminate the mutual feedback between cells of the TME and carcinoma cells. We have now provided additional experiments and clarified the novelty of our study in the revised text.

3. To contribute to cancer research, the scRNAseq data would need to be uploaded in a platform that can all users have access (e.g. single cell portal from broadinstitute).

Thanks for this comment. All of our raw data can be publicly accessible through the gene expression omnibus (GSE203608) upon publication of this study. Data that is compatible with VisCello has also been deposited to Zenodo (<https://zenodo.org/record/7872684>). Readers can utilize VisCello (<https://github.com/qinzhu/VisCello>) to interactively explore and analyze the deposited data.

4. Liver metastases results are unclear. I understand the limitations to work with liver metastases but there are only 4 samples, most of the sample is non-epithelial and number of cells analysed are less than 6,000. Most of the findings are described in supplementary but I consider that may not be significant. I suggest authors to think about how significant these results are.

We agree that the small number of samples makes it difficult to draw definitive conclusions regarding metastatic lesions, however we felt that including this data in the supplement may provide a useful reference for other groups analyzing human colon cancer metastases, as large datasets from matched primary tumors and metastatic lesions of colorectal adenocarcinoma are not yet available.

Despite this limitation we have observed consistent changes in macrophage state (Fig. 4F), T and B cell composition (Fig. S8C, S8F and Fig. S9C) in metastatic sites compared to the primary tumor, suggesting carcinoma cells may also remodel the TME in the liver, perhaps through the induction of the SPP1+ macrophage state to induce immunosuppression. We have carefully examined the language related to this data to insure we are not overstating the results and to highlight the limitations of the small sample size.

5. The findings shown in the figures are vague described in the text. It many times says " most of the pathways..", "many suppressed..." "found a striking suppression of gene expression programs.." but does not specify numbers, %, significance, statistics, or any details about the findings.

Thanks for this, we have added specific quantitative values, primarily in the figures and legends to convey the results of the analyses more precisely.

6. There can be a different TME depending on the stage of tumour, can this be identified by the scRNAseq analysis? or is it needed a higher number of samples?

As the reviewer requested, we added stage information as well as MSS/MSI information to our analysis. In the revised Fig. 2B, which reflects the average composition of the TME, significant increases in certain TME cell types can be observed in later stages. When looking at the macrophage compartment, we observe that later stages (II, III, IV) exhibit significant enrichment of SPP1+ macrophages relative to stage I, and macrophages in the C1QC+ state are significantly reduced. We also investigated the proportions of T cell subtypes across tumor stages (Fig. S8E) and found increase in the regulatory T cell population (associated with immunosuppression) and decrease in the T effector/T effector memory cells (associated with antitumor immune response) in tumors relative to normal tissue.

Minor

comments:

1. Figure 1c, it should be written colonoid and tumoroid as has been done in the text.

We have revised the text based on the reviewer's suggestion.

2. Figure 1b, tumours could be also ordered following stages to clarify the type of samples, and figure 1d can also be ordered from left to right by stages. Can the authors include images of all CRC organoids? in figure 1d, the organoids phenotype can be hardly seen.

Thanks for pointing this out- we've now provided higher resolution/higher magnification images for Fig. 1D so gross organoid morphology is better appreciated, and we include images of all organoid cultures for which we present single cell transcriptomic data. We have also re-ordered the patient samples in Fig. 1B by stage, as suggested.

3. It is written "for at least 4 passages", but nothing about maximum of passages done in these two months.

We have now clarified that organoids are passaged 4-6 times prior to any analyses.

4. Figure 2c, it cannot be seen the stainings of each marker as the magnification is low and the individual stainings are not shown. This happens with some other stainings throughout the text.

Thanks, we apologize for this, and have now included higher magnification images and panels with individual channels (revised Fig. S2).

5. This sentence should be re-written: "To this end, we asked how well human tumoroids or various mouse models (including the Apcmin model of familial adenomatous polyposis, the AOM-DSS model of inflammation-driven colorectal cancer, and the APKS model of invasive, metastatic colorectal adenocarcinoma generated by endoscope-guided orthotopic implantation of

CRISPR/Cas9-engineered tumoroids with Apc, p53, Kras, and Smad4 mutations into the colonic mucosa of syngeneic mice) recapitulate human primary tumors and tumoroid model tumoroids or various models" It is long and not clear for the reader.

Thanks for pointing this out, we have broken this sentence down into several simple sentences for ease of understanding.

Reviewer #2 (Remarks to the Author): Expert in single-cell and spatial omics, colorectal cancer and tumour immune microenvironment.

Colorectal cancer (CRC) is one of the most common malignant tumors worldwide and the global burden and lack of major therapeutic advances in CRC give rise to high mortality rates. Mouse tumor model plays a very important role in the research of colorectal cancer, but invasive adenocarcinoma and gene editing will result in the premature mortality of the model. Recently, the development of human organoids enables the modeling of a range of immune checkpoint blockades, but the co-culture of organoids has difficulty reproducing the interaction between tumor cells and the tumor microenvironment.

In this manuscript, the authors collected 17 CRC patients and conducted single-cell RNA sequencing with the primary tumor, normal adjacent tissues, metastatic lesions. Also, the authors conducted single cell RNA sequencing with normal organoids and tumoroids cultured from human epithelium, which is very valuable data. The authors compared the advantages and limitations between mouse model and organoids in mapping and modeling human colorectal carcinoma interactions with the tumor microenvironment, which will be helpful for selecting efficient methods in oncology research. However, it has been reported that SPP1+ Macrophage is highly related to the progression of tumor, the innovation is insufficient in biological meaning. Also, the authors need to address the following concerns:

We thank the reviewers for their insightful comments which we have addressed point-by-point below. In the revised manuscript we have also provided further analysis and insight into the genesis of SPP1+ macrophage polarization, and how this relates to the immunosuppressive microenvironment, building upon the observation in the literature that SPP1+ macrophages correlate with tumor progression, as noted by the reviewer.

Major

comments:

1. In Figure 3B and 3D, primary normal adjacent epitheliums are more similar to mature absorptive colonocytes and primary tumor are more similar to crypt base columnar stem cells. Human normal and tumor epitheliums have significant difference in transcriptional identities, while the normal organoids and tumoroids both shift towards stem cell. Have the organoids lost the characteristics of human normal and tumor tissues during the *in vitro* culture? The authors should show more evidence to validate the correlation between organoids and the human epithelial samples.

We observe that the most well-correlated samples are the normal adjacent (non-tumor) samples. This correlation is strong both across primary patient-derived samples as well as across the

normal colonoids. In Fig. 3B we can see most of the Pearson correlations are above **0.8**, including between the *in vivo* and *in vitro* samples, indicating that most of the transcriptome characteristics are retained after *in vitro* culturing. This is further demonstrated in Fig. 3G, where DE analysis showed that many genes up-regulated in carcinoma cells are also up-regulated *in vitro*, and thus many tumor cell signatures are maintained *in vitro*. These correlations are decreased in tumors across patients, perhaps not surprisingly as these tumors are likely driven by divergent mutations and represent different stages of tumor ontogeny.

Ultimately, to speak to the reviewers point as to whether ‘organoids have lost the characteristics of human normal and tumor tissues during *in vitro* culture’, this is a central point to our study: Changes *do* occur when epithelial cells are removed from the patient and are maintained in prolonged organoid culture. These changes are primarily a suppression of gene expression programs related to inflammation and communication with the immune system in tumoroid cultures, as well as an increase in proliferation-associated signatures. This is also reflected by changes at cell state level, where organoids and tumoroids exhibit a shift in distribution towards stem cell, transient amplifying cell, and enteroendocrine cells (EEC) states, while colonocytes (CC), goblet cells (G), Paneth like cells (PLC) states show significant decreases in tumoroids. We have now included additional analysis of cell state distributions across samples in revised Fig. S5. These changes are consistent with the biology of the system: no immune cells are present in the long-term organoid cultures, and these cultures are nutrient- and cytokine-replete. The latter explains the observed shift toward enhanced stem and progenitor cell frequency at the expense of differentiated cell types (a well-established paradigm in the literature). Beyond these shifts in the balance of cell type abundance and loss of communication with the TME, we find no evidence that cultured cells have lost characteristics found in their *in vivo* counterparts.

2. In addition to macrophages, the T and B cells are also play important roles in the epithelium-immune interaction. In Figure 5A, the authors tried to model the human CRC TME in mouse model, but there are few T and B cells. More mouse samples may be helpful to increase the proportion of T and B cells. And the authors need to supplement the detail and metadata of mouse model scRNAseq to the methods.

We thank the reviewers for bringing up this important aspect of TME, which we did not include in our initial analysis. T and B cells have been clearly shown to play important roles in carcinoma initiation and progression, and this has been extensively examined in a prior study applying single cell transcriptomics to human colorectal cancer (PMID: **30479382**). We did not focus on the adaptive immune system in this study both for this reason, and because we cannot perform functional experiments testing B cell or T cell interactions with carcinoma cells in organoids as we do not have access to these syngeneic cells from patients (and thus there will be MHC mismatch along with a lack of T cells with specificity for a tumor antigen). To the point that there are few T and B cells in the mouse data, the proportion of B/T cells in the mouse datasets is generally consistent with the proportions observed in the human patient samples, and it is unclear how increasing the number of samples would increase the proportion of these adaptive immune cells in the absence of positive selection for them (e.g., by MACS or FACS), which we believe to be

out of the scope of the current study, given that our focus is on carcinoma cells, the effects of organoid culture on their identity, and the reconstruction of macrophage-carcinoma interactions.

Despite these limitations, the number of T and B cells in the human data are sufficient for us to identify key T and B cell states, now presented in (Fig. S8 and Fig. S9) where we performed in-depth analysis on these adaptive immune cell populations. We identified major T/NK cell subpopulations including naive or memory T cells (Tn/Tm, CCR7+), regulatory T cells (Tregs, FOXP3+), effector T or effector memory T cells (Teff/Tem, GZMA/GZMB+), short-lived effector cells (SLEC, KLRG1+), NKT cells (CD3+NKG7+) as well as a population of cycling T cells (MKI67+), a cluster of T-carcinoma cell doublets, and NK cells. The distribution of these T/NK cell subpopulations differs significantly across sample types (Fig. S8C, 8D), with tumor and liver metastasis samples showing some striking commonality, despite of the different anatomical location.

Specifically, we observed enrichment of Tregs, Teff/Tem, and reduction of Tn/Tm and NK cells in tumor tissue relative to normal colon/liver (Fig. S8D). There is also increase in Treg proportion and decrease in the Teff/Tem population in late-stage patients (Fig. S8E), indicative of a more immunosuppressive environment. For almost all T cell subpopulations, the exhaustion score, computed based on a set of known T cell exhaustion markers, exhibited significant elevation when comparing T cells in the TME against those in adjacent normal tissue (Fig. S8F).

Lastly, we asked if there exists an association between the observed macrophage states and T cell states by looking at the correlation between the fraction of the T cell subpopulations in each sample, and the average IL1B/SPP1/C1QC state probability in macrophages. As shown in Fig. 4M, 4N and Fig. S8G, the IL1B+ macrophage state is highly correlated with the proportion of Tn/Tm and NK cells. In contrast, the SPP1+ state is positively correlated with the proportion of Treg, SLEC and NKT cell, and negatively correlated with the proportion of Tn/Tm, Teff/Tem, NK cell and cycling T cells. C1QC+ state probability shows almost the opposite trend. The trend is even more striking if only tumor samples are included in the analysis (Fig. 4N, Fig. S8G). These observations are highly consistent with the observations in our study and in the broader literature that the SPP1+ macrophage state is indicative of a strong immunosuppressive TME. It further indicates that T cell states and macrophage states are highly coordinated, possibly due to the crosstalk between the two cell types.

A set of similar analyses were performed on the B/plasma cell subpopulations. We were able to identify naïve B cells, a group of cycling/differentiating B/plasma cells, and different plasma cell populations expressing different antibodies. In the normal colon, B cells are mainly present in the form of IGHA-expressing plasma cells involved in maintaining intestinal immunity. In most cancer patients, there is a switch to IGHG-expression plasma cells (Fig. S9C, D). Interestingly, we observed that in one patient (P87), the plasma cells are almost all IGHA+, but with a significant shift in transcriptome compared to that observed in normal colon. When computing the correlation with macrophage state probability, strong suppression of B cell proliferation and differentiation was observed in samples with high SPP1 signature (Pearson correlation = -0.37), and the inverse

trend was seen for the C1QC+ state (Pearson correlation = 0.47), suggesting the different roles these macrophages may play in the tumor TME (Fig. S9E, F).

3. In Figure 6C, the authors claimed that there is a clear shift in macrophage polarization toward the SPP1+ state. Whether this phenomenon can be observed in P08 and P24? And the normal organoids and tumoroids seem to have a similar influence on macrophages during the co-culture.

We now include the analogous macrophage-carcinoma coculture experiments with lines P08 and P24 as requested and observe a similar shift towards SPP1+ macrophage state in these co-culture experiments. In these assays we also included an additional condition- the addition of cancer-associated fibroblasts and present this data in the revised Fig. S10.

Minor

comments:

1. In Figures 2A and 5A, the authors need to check the small populations of cells far away from the other cluster, and filter them if they are contaminated.

Most of these small clusters are epithelial cell clusters which represent individual epithelial cell subtypes/states including goblet, enteroendocrine, or Paneth-like cells (for which we now provide more in-depth analysis in Fig. S4), or carcinoma cells from different patients separated because of transcriptome heterogeneity across patients. Other small clusters which are not epithelial cells have been identified and annotated as TME cell types present at low abundance, such as hepatocytes or mast cells. They are not artifacts or contamination. Therefore, we believe that retaining these cells in the UMAP gives reader a more complete picture of the epithelial cell states and TME composition.

2. In Figure 2D, the y-axis may not be the cell type. As the author described, it may be the patient's ID.

Perhaps the reviewer is referring to supplemental Fig. 2D, where the Y-axis is the Patient ID and sample type, and the X-axis is cell type. We apologize, the text in the figure legend is unclear and we have corrected it, thanks for catching this!

Reviewer #3 (Remarks to the Author): Expert in colorectal cancer clinical and molecular research, and tumour-associated macrophages

Tumor-associated macrophages (TAMs) constitute the dominant immune cell population in colorectal cancer (CRC). Recent studies have shown that some of these cells are continuously recruited from the blood and, upon interaction with factors released in the tumor microenvironment

(TME), express specific genes, such as SPP1, and become immunosuppressive, thus contributing to generate a niche that sustains cancer cell growth and diffusion. By using single-cell transcriptomic analyses of treatment-naïve, surgically resected human CRC samples, Li and colleagues sought to map putative interactions between human CRC cells and TAMs. In parallel, by establishing CRC organoids (tumoroids) and normal colon organoids, the authors assessed the effect of cancer cells on TAMs differentiation. As expected, primary tumors contained a variety of cell types, including immune and non-immune cells. Next, by using CODEX analyses, the authors showed that carcinoma cells had a higher propensity for interaction with stromal cells and immune cells relative to their normal counterparts, and documented potential receptor-ligand interactions between carcinoma cells and cells within their microenvironment. It was also shown that the average transcriptome of cells from tumoroids and organoids was quite similar while there were differences between primary tumors and the derived tumoroids. In this context, several genes were suppressed and others were activated in tumoroids as compared to primary carcinoma cells. Specifically, genes involved in communication with immune cells were suppressed while those related to cell division, patterning, and metabolism were generally activated in tumoroids consistent with the shift to stem and progenitor cell states and away from terminally differentiated absorptive states in culture. The latter findings were then seen in various mouse models of colon tumorigenesis. In the last parts of the study, the authors focused on the distribution of normal and tumor macrophages and showed that macrophages in the SPP1+ state, which is known to suppress anti-tumor immune responses, are more prevalent in tumors relative to the other states. Consistently, stratification of colorectal cancer TCGA transcriptomes by SPP1 signature enrichment revealed poor survival in patients exhibiting high SPP1 signature. SPP1 itself encodes osteopontin, a secreted ECM component, and ligand for CD44, the latter being up-regulated in tumors relative to normal tissue. Similar findings were seen in all the mouse tumor models. Finally, culturing circulating monocyte-derived macrophages with tumoroids induced a shift in macrophage polarization towards the SPP1+ state.

Although the topic is of interest and this reviewer acknowledges the enormous amount of work performed by the authors to analyze the single-cell transcriptomic data, the manuscript is largely descriptive and several issues/points were not adequately taken into consideration by the authors. Specifically, the main conclusion of the study is a little bit skewed to convince the reader that differentiation of suppressive macrophages is exclusively driven by cancer cells.

We thank the reviewer for their insightful and thorough comments, which we address individually below. To the main above point that our conclusion is 'skewed to convince the reader that differentiation of suppressive macrophages is exclusively driven by cancer cells', we do not intend to suggest that carcinoma cells can exclusively do this, as the literature has implicated other drivers of this phenotype, including mesenchymal cells, the ECM, and hypoxia, which we discuss and cite in the manuscript.

We have altered the language to make it clear that the influence of carcinoma cells themselves is contributory to the SPP1+ immunosuppressive phenotype but is not necessarily the major or only driving force. We have also now included additional experimental data adding cancer-associated fibroblasts with or without macrophages to tumoroid culture to address the relative

contributions of these different cell types to macrophage polarization, described in more detail below.

Specific points

1. The elevated expression of SPP1 has been previously identified among TAMs across several cancers including primary CRC (*Cell* 2021;184(3):792-809 e23). The innovativeness of the manuscript is also partly hampered by a recent study showing that metastatic CRC express predominantly SPP1+ macrophages and these communicate with fibroblasts with the downstream effect of promoting immunosuppression in the TME (*Clin Cancer Res.* 2022 Oct 14;CCR-22-2041).

We agree with the review that the correlation between SPP1+ macrophages, an immunosuppressive microenvironment, and more aggressive tumor growth have been reported in the literature, including in the recent high-profile *Cell* and *Cancer Cell* studies within the past 1-2 years mentioned above. The focus and novelty of our current study lies in our evaluation of how organoid culture alters the identity of carcinoma cells, the finding that gene expression patterns associated with communication between carcinoma and immune cells are lost in organoid culture, and the finding that some of this molecular crosstalk can be re-established via the introduction of innate immune cells into the organoid culture. We do not make any claims of novelty surrounding the identification of SPP1+ macrophages and have cited prior literature, indicating that our current study adds further support for these prior studies which implicate SPP1 in CRC metastasis from scRNAseq surveys.

Further, we now provide additional analyses suggesting that the abundance of SPP1+ macrophages is positively correlated with the proportion of suppressive T cell subtypes such as regulatory T cells, and negatively correlated with effector T or effector memory T cells. A similar effect can be seen on the B and plasma cells, where differentiation and proliferation of B cells are significantly reduced in patients with a high abundance of SPP1+ macrophages. These provide further evidence that the SPP1+ macrophages are associated with immunosuppressive TME. In addition, our original and newly added co-culture experiments demonstrated that the SPP1+ macrophage state can be induced by epithelial cells alone, while coculture with cancer-associated fibroblasts (CAFs) failed to polarize macrophages into the SPP1+ state *in vitro*. To our knowledge this data is novel and has not been described in the published literature which, in the context of colon cancer, has primarily relied on scRNAseq data to draw correlations between SPP1+ macrophages and tumor progression rather than the execution of functional experiments.

2. A major finding of the study is that cancer cells promote the differentiation of suppressive macrophages, which sustain tumorigenesis. No attempt was made by the authors to examine whether cancer cells either release factors that induce SPP1+ macrophages or such a differentiation relies on cell-cell interactions.

This is a good point, and to address it we have cultured naïve human monocyte-derived macrophages with conditioned media from organoids to determine if the inducing factor(s) are soluble, or whether direct cell-cell interaction is required. As shown in Fig. S10 of the revised manuscript, conditioned media from tumoroids and organoids cause macrophages to differentiate

towards the C1QC+ state which expresses high level of MHC Class II genes, indicating that direct cell-cell contact plays a key role in suppressing this fate and promoting the immunosuppressive SPP1+ fate.

3. Although the authors documented potential receptor-ligand interactions between carcinoma cells and cells within their microenvironment, none of such interactions was eventually validated.

We did not prioritize functional validation of these interactions as they have been validated in prior literature (which resulted in the curation of receptor-ligand interaction databases used to discover these interactions). In the revised manuscript we did, however, seek to validate the key interaction between Osteopontin (encoded by *SPP1*) and its predicted receptor CD44. New proximity ligation assays confirm the interaction predicted from the scRNAseq data in macrophage-tumoroid cocultures and are presented in the revised Fig. 6J. We also confirmed that SPP1+ macrophages can be found in close proximity to carcinoma cells *in vivo* using single molecule RNA FISH assays (as Osteopontin is a secreted molecule), presented in Fig. 4L.

4. The *in vitro* studies do not help understand if the SPP1 phenotype induced by culturing tumoroids with circulating monocyte-derived macrophages is stable over time. Moreover, no mechanistic study was performed to assess the suppressive functions of the *in vitro* differentiated SPP1+macrophages.

Unfortunately, the first point regarding the stability of the SPP1+ phenotype in macrophages over time cannot be addressed in this system as the culture conditions which promote long-term tumoroid growth are incompatible with long-term macrophage survival. In fact, human monocyte-derived macrophages cannot be stably cultured in any conditions- to the best of our knowledge the longest experiments conducted with these cells *in vitro* is only around one week. Ultimately, the question the reviewer may be getting at is whether the induction of the SPP1+ state is epigenetically wired into the macrophages such that they retain this state even if the inductive signal is no longer present. This is a fascinating question; however, it would require an entirely new study.

With regard to the second point, we demonstrate that macrophage co-culture with tumor organoids not only promotes induction of the SPP1 state in macrophages, but that the carcinoma cells respond reciprocally via induction of mesenchymal markers and EMT gene signatures, consistent with prior reports of SPP1 function in other carcinoma types (doi.org/10.1155/2021/5806602, doi.org/10.3389/fcell.2021.646390, doi.org/10.1038/s41389-020-00262-2.)

Our findings suggest that SPP1+ macrophages similarly promote EMT in human colon cancer. The notion that SPP1 is an immunosuppressive molecule which supports carcinoma metastasis is reported in the literature (doi.org/10.1158/0008-5472.CAN-13-3334, doi.org/10.1016/j.tcb.2005.12.005), including a demonstration that SPP1/CD44 interaction inhibits T-cell activation to facilitate tumor immune evasion (doi.org/10.1172/JCI123360). In light of this literature, and the focus of our study on how organoid culture influences carcinoma cell

state and how carcinoma cells influence macrophage polarization, we have not performed additional functional experiments to further confirm the published immunosuppressive effects of SPP1+ macrophages. Importantly, it is experimentally nearly impossible to test T cell killing in our experimental model. First, we don't have T cells from the same patients that the tumoroids were derived from, and as such there would be MHC mismatch between individuals. Secondly, to test T cell killing, one needs a population of T cells with specificity for a particular antigen. This can be done in mice with the OT/Ovalbumin system, where transgenic mice overexpressing an ectopic antigen from ovalbumin are used to derive T cells and test their ability to kill tumor cells expressing this antigen. In the human culture models used in our study, such tools do not exist.

Despite the difficulty in experimentally assessing the immunosuppressive functions of SPP1+ macrophages *in vitro*, our new bioinformatics analysis of the T cell and B cell subpopulations (Fig. S8 and S9), as described above, clearly demonstrate a strong association between SPP1+ macrophages and immunosuppressive states of adaptive immune cells *in vivo*. Combined with our observation that carcinoma cells alone can induce the SPP1+ macrophage state, and suppress the MHC II presenting C1QC+ state *in vitro*, it would be reasonable to hypothesize that carcinoma cells may induce immunosuppression at least in part by contributing to SPP1+ macrophage polarization. Our results provide key insights for future mechanistic studies on this topic.

5. Accumulating evidence indicates that differentiation and function of TAMs are mainly controlled by non-immune cells present in the TME (i.e. fibroblasts). Therefore, the *in vitro* studies should be implemented with further research aimed at evaluating the effect of cancer-associated fibroblasts (CAFs) on the differentiation of SPP1+ macrophages. Additionally, the authors should better characterize the different stromal cells present in the CRC microenvironment as well as the ability of SPP1+ macrophages and CAFs to influence the tumor-associated T cells.

We thank the reviewer for these great suggestions. As suggested, we have now conducted an additional series of experiments in which human macrophages are cultured with or without tumoroids, and with or without the patient-derived colorectal cancer-associated fibroblasts.

In the new co-culture experiment, shown in Fig. S10, we found CAFs alone do not robustly induce the SPP1+ state. Further, while the state can be induced with organoid/tumoroid cells alone, conditioned media from these cells is insufficient. These observations indicate that direct cell-cell contact between epithelial cells and the macrophages may be critical for induction of the SPP1+ state. Intriguingly, when macrophages were co-cultured with both epithelial cells and CAFs, they exhibited further shift towards the SPP1+ state, suggesting that CAFs may facilitate induction and stabilization of this state.

We also added new analysis on the CAF populations from our *in vivo* data as the reviewer suggested (Fig. S6). The analysis shows both fibroblasts and myofibroblasts are highly abundant in the TME in both primary site and liver mets site, and the fibroblasts in tumor samples up-regulate several ECM genes known to promote tumor progression and metastasis, including *CTHRC1*, *BGN*, *INHBA*, and *PDPN*.

Regarding functionally assessing T cell killing, as described above in the response to point 4, it is not technically feasible to assess the ability of SPP1+ macrophages to influence T cell killing in this experimental paradigm. However, it is well established in the literature that SPP1/Osteopontin interaction with CD44 inhibits T cell activation and promotes immune evasion by the tumor (see, for example PMID: **30395540**), and we now provide evidence using proximity ligation assays in macrophage-tumoroid cocultures that CD44 and Osteopontin interact in this system.

6. Do normal colon organoids promote differentiation of C1QC+ monocyte-derived macrophages? Is such a differentiation abolished/shifted by CAFs?

No, in both our previous and new co-culture experiments, there is moderate suppression of the C1QC+ state by normal organoids from most patients. The further addition of CAFs did not have any effect on the C1QC+ state (Fig. S10).

7. CRCs with high-level microsatellite instability (MSI) are marked by increased infiltration of T cells in the TEM. In contrast, CRCs with microsatellite stable (MMS) generally do not have significant levels of T cell infiltration and have a profoundly immunosuppressed TME (Hirano et al. 2021 Jap J Clinical Oncology, 51: 1, p. 10–19; Immunity 2016;44:698–711). The authors should assess whether the prevalent accumulation of SPP1+macrophages is restricted to CRC with MMS.

Thanks for this suggestion. While this study focuses on MSS patients, we have a few MSI patients, as now shown in Fig. 4G, SPP1+ macrophages are significantly enriched in MSS patients compared to MSI patients and are further positively associated with tumor stages (Fig. 4F).

8. The authors stated that they were able to capture a total of 38,063 cells from the 15 primary tumors. Figure 1E shows the enormous variability among the tumor samples, and the number of cancer cells recovered from more than fifty percent of samples was very low (less than 2000) or absent. These findings, together with the lack of information regarding the procedure adopted to eliminate cells expressing genes indicative of cell death, introduce bias in the interpretation of the results.

The variability in the number of cancer cells is directly related to the time elapsed between surgical resection and processing, as epithelial cells are much more prone to cell death upon dissociation relative to non-epithelial cells. Because of this, we first apply a strict filter to the dataset to exclude low-quality cells or debris, requiring a cell must at least have 1000 transcripts (UMIs) and 500 expressed genes to be included. Furthermore, cells are filtered based on the fraction of transcripts emanating from mitochondria genes, where higher fractions are indicative of cell death. Therefore, for all analysis, cells with greater than 20% mitochondrially derived

reads were excluded. Several tumor samples with too few epithelial cells were excluded from the epithelial-cell-specific analysis and were only included when analyzing the TME. We now include this detailed information in the manuscript.

9. Markers identified from the scRNA-seq results should be confirmed by protein analysis (i.e. immunostaining).

Beyond the CODEX immunoanalysis, which provides a protein-level readout of cell type distribution and a spatial correlate the single cell transcriptomics, we know also include a proximity ligation assay for detection of interaction between key molecules involved in immunosuppression, Osteopontin and CD44. Finally, because Osteopontin is a secreted ECM component, we include additional single molecule FISH images depicting *SPP1* transcripts in macrophages in tumor tissue.

10. I could not find the suppl Tables indicated in the text.

Sorry, we now include these supplemental tables.

REVIEWERS' COMMENTS

Reviewer #1 (Remarks to the Author):

The manuscript has been improved. I still think that is very descriptive but still deserves its publication in Nature Communications. The information provided will advance TME field.

Reviewer #2 (Remarks to the Author):

The authors have substantially improved their manuscript. They emphasized that the novelty of the manuscript lies in whether "organoids" lose similar characteristics to human normal and tumor tissues during in vitro culture. They also additionally analyzed T cells and B cells, and the results showed that the T cell state and the macrophage state were highly coordinated, possibly due to the crosstalk between two cell types.

There is only one minor comment here. Since the authors mentioned that the small groups in Fig. 2A were meaningful, please show whether they come from one sample or exist in multiple samples.

Reviewer #3 (Remarks to the Author):

I appreciated the authors' effort to revise their manuscript according to the reviewers' suggestions. The overall data are now convincing and support the authors' conclusions. Congratulations to the authors for this excellent work.

Reviewer #1 (Remarks to the Author):

The manuscript has been improved. I still think that is very descriptive but still deserves its publication in Nature Communications. The information provided will advance TME field.

-We thank the reviewer for their thoughtful comments which have contributed greatly to improving the final manuscript.

Reviewer #2 (Remarks to the Author):

The authors have substantially improved their manuscript. They emphasized that the novelty of the manuscript lies in whether "organoids" lose similar characteristics to human normal and tumor tissues during in vitro culture. They also additionally analyzed T cells and B cells, and the results showed that the T cell state and the macrophage state were highly coordinated, possibly due to the crosstalk between two cell types.

There is only one minor comment here. Since the authors mentioned that the small groups in Fig. 2A were meaningful, please show whether they come from one sample or exist in multiple samples.

-We thank the Reviewer for their feedback on the revised study and have addressed their final point. We have provided an additional supplemental figure panel that speaks to this point (Supplemental Figure 2F&G). These small UMAP clusters were primarily carcinoma cells coming from different patients. This is unsurprising given that the cancers between patients harbor different mutational burdens, which translates to differences in transcriptional states observed in UMAP space. This can also be appreciated in Figure 3A where we look only at the epithelial/carcinoma fraction in the dataset.

Reviewer #3 (Remarks to the Author):

I appreciated the authors' effort to revise their manuscript according to the reviewers' suggestions. The overall data are now convincing and support the authors' conclusions. Congratulations to the authors for this excellent work.

-We thank the reviewer for their thoughtful comments and support of our study!